# Yap regulates skeletal muscle fatty acid oxidation and adiposity in metabolic disease

K. I. Watt[1,2,3,4], D. C. Henstridge [5], M. Ziemann[6], C. B. Sim[7], M. K. Montgomery[2], D. Samocha-Bonet [8,9], B. L. Parker[1,2], G. T. Dodd [2], S. T. Bond [3], T. M. Salmi[10,11,12], R. S. Lee[13], R. E. Thomson[1], A. Hagg[1], J. R. Davey[1], H. Qian[1], R. Koopman[1], A. El-Osta [4,14,15], J. R. Greenfield[8,9,16], M. J. Watt[2], M. A. Febbraio [17], B. G. Drew [3], A. G. Cox [10,11,12], E. R. Porrello [2,7], K. F. Harvey [10,12,18] & P. Gregorevic[1,2,3,19]✉

Obesity is a major risk factor underlying the development of metabolic disease and a growing public health concern globally. Strategies to promote skeletal muscle metabolism can be effective to limit the progression of metabolic disease. Here, we demonstrate that the levels of the Hippo pathway transcriptional co-activator YAP are decreased in muscle biopsies from obese, insulin-resistant humans and mice. Targeted disruption of Yap in adult skeletal muscle resulted in incomplete oxidation of fatty acids and lipotoxicity. Integrated 'omics analysis from isolated adult muscle nuclei revealed that Yap regulates a transcriptional profile associated with metabolic substrate utilisation. In line with these findings, increasing Yap abundance in the striated muscle of obese (*db/db*) mice enhanced energy expenditure and attenuated adiposity. Our results demonstrate a vital role for Yap as a mediator of skeletal muscle metabolism. Strategies to enhance Yap activity in skeletal muscle warrant consideration as part of comprehensive approaches to treat metabolic disease.

[1] Centre for Muscle Research, The University of Melbourne, Melbourne, VIC, Australia. [2] Dept of Physiology, The University of Melbourne, Melbourne, VIC, Australia. [3] Baker Heart and Diabetes Institute, Melbourne, VIC, Australia. [4] Dept of Diabetes, Central Clinical School, Monash University, Melbourne, VIC, Australia. [5] School of Health Sciences, University of Tasmania, Hobart, Tas, Australia. [6] Deakin University, Melbourne, VIC, Australia. [7] Murdoch Children's Research Institute, Melbourne, VIC, Australia. [8] Division of Healthy Aging, Garvan Institute of Medical Research, Darlinghurst, NSW, Australia. [9] St Vincent's Clinical School, Faculty of Medicine, University of New South Wales, Sydney, NSW, Australia. [10] Peter MacCallum Cancer Centre, Melbourne, VIC, Australia. [11] Dept of Biochemistry and Molecular Biology, The University of Melbourne, Melbourne, VIC, Australia. [12] Sir Peter MacCallum Dept of Oncology, The University of Melbourne, Melbourne, VIC, Australia. [13] Metabolic Disease and Obesity Phenotyping Facility, Monash University, Melbourne, VIC, Australia. [14] Dept of Pathology, The University of Melbourne, Melbourne, VIC, Australia. [15] Hong Kong Institute of Diabetes and Obesity, Prince of Wales Hospital, The Chinese University of Hong Kong, Shatin, Hong Kong. [16] Dept of Diabetes and Endocrinology, St Vincent's Hospital, Darlinghurst, NSW, Australia. [17] Drug Discovery Biology, Monash Institute of Pharmaceutical Sciences, Monash University, Melbourne, VIC, Australia. [18] Dept of Anatomy and Developmental Biology, and Biomedicine Discovery Institute, Monash University, Melbourne, VIC, Australia. [19] Dept of Neurology, The University of Washington School of Medicine, Seattle, WA, USA. ✉email: pgre@unimelb.edu.au

Obesity is a major healthcare issue affecting more than 600 million individuals world-wide, with 350 million individuals currently suffering from obesity-driven type 2 diabetes and associated diabetic complications[1,2]. Obese humans and mice demonstrate features of skeletal muscle dysfunction including muscle fibre atrophy, insulin resistance and altered substrate metabolism[3–5]. Importantly, stimulating skeletal muscle metabolism via interventions such as physical activity is associated with improved whole-body metabolic capacity[6,7]. However, adherence to the required amounts of physical activity in obese individuals remains low[8]. The rising incidence of obesity-driven pathologies highlights the urgent requirement to develop effective interventions to treat metabolic dysfunction. A deeper understanding of the signalling networks that impact skeletal muscle metabolism, and their contribution to metabolic dysfunction, may help to identify new pharmacological and lifestyle interventions that can be exploited to combat metabolic disease.

The Hippo signalling pathway is a critical mediator of organ development across species that functions by limiting the activity of the transcriptional co-activators Yap and Taz[9,10]. Yap and Taz, and their *Drosophila* orthologue Yorkie, shuttle dynamically between the cytoplasm and nucleus[11–13]. When nuclear, Yap and Taz can physically associate with the TEA-domain (Tead) transcription factors to control transcriptional programmes that influence cell fate, metabolism, and survival[12,14–17]. In skeletal muscle, Yap-Tead and Taz-Tead complexes operate as important post-mitotic regulators of cell size[18–21]. Activation of Yap-Tead and Taz-Tead complexes in adult mammalian muscles can enhance skeletal muscle size[18,19,22]. Conversely, inhibition of Yap activity in adult mammalian skeletal musculature can cause a reduction in muscle fibre volume by inhibiting anabolic processes[19]. Furthermore, changes in Hippo pathway activity have been reported in muscles undergoing atrophy associated with aging, disruption of the neuromuscular junction, or corticosteroid administration, and in muscles undergoing hypertrophy associated with increased loading[18–20,23,24]. Combined, these observations identify the transcriptional regulators Yap and Taz as important post-natal effectors of skeletal muscle biology in health and disease, yet the biological processes controlled by the Hippo pathway in skeletal muscle remain unclear.

Here, we demonstrate that Yap regulates fatty acid oxidation and a transcriptional programme associated with metabolic substrate utilisation in adult skeletal muscle. YAP levels are reduced in the muscles of obese insulin-resistant humans and mice, with knockdown of Yap in mouse limb muscles resulting in incomplete oxidation of fatty acids. Yap knockdown impacts a programme of gene expression associated with metabolism that includes the TCA cycle enzyme, Idh2. Importantly, increasing Yap levels in the muscles of obese *db/db* mice attenuates adiposity by increased energy expenditure, independent of changes in lean mass, food intake or activity. Collectively, these findings provide evidence for an important role of the Hippo pathway as a post-natal regulator of metabolism in skeletal musculature.

## Results

### Yap abundance is reduced in the skeletal muscles of insulin-resistant humans and mice.

To explore a potential metabolic role for the Hippo pathway in skeletal muscle, we first sought to establish if the abundance of YAP, the core effector of the Hippo pathway, was altered in the muscles of insulin-resistant humans. YAP levels were assessed in vastus lateralis muscle biopsies obtained from obese, non-diabetic humans that were stratified according to skeletal muscle insulin sensitivity following a hyperinsulinemic-euglycemic clamp (Fig. 1A). Consistent with the hypothesis that YAP functions as a positive mediator of

metabolism in skeletal muscle, we found that YAP protein levels were reduced in the muscles of obese, insulin-resistant individuals compared to obese, insulin-sensitive individuals (Fig. 1B, Supplementary Fig. 1A). We also identified a positive correlation between YAP levels in human muscle biopsies and the glucose infusion rate relative to fat free mass (FFM) during a hyperinsulinemic-euglycemic clamp; a gold standard measure of skeletal muscle insulin sensitivity (Fig. 1C). Further, we observed a positive correlation between skeletal muscle YAP levels and the difference in respiratory quotient (RQ) from fasted to glucose-infused states ($\Delta$RQ), demonstrating impaired whole-body metabolic flexibility is associated with reduced YAP abundance in the skeletal muscle of obese humans (Fig. 1D).

We next assessed the levels of Yap in the skeletal muscles of mouse models of genetically-induced hyperphagic obesity (GIO; leptin receptor null; *db/db* strain[25]) and diet-induced obesity (DIO; 6 weeks 43% high-fat diet feeding of C57BL/6J). Consistent with our observations in insulin-resistant humans, Yap levels were markedly reduced in the predominantly glycolytic tibialis anterior (TA) muscles of *db/db* mice at 6 weeks of age compared to littermate *db/+* controls (Fig. 1E–G). In contrast, we found that Yap levels were increased in the predominately oxidative *soleus* muscle (Fig. 1F, H, Supplementary Fig. 1B, C). No differences were observed in *Yap* mRNA expression between *db/db* and *db/+* TA muscles indicating that the reduced levels of Yap protein were due to post-transcriptional mechanisms (Supplementary Fig. 2A). The abundance of Yap was similar in the TA and Soleus muscles of mice fed standard lab and high-fat diets for 6 weeks (Supplementary Fig. 2B). Collectively, these observations demonstrate dysregulation of the Hippo pathway in the skeletal muscles of insulin-resistant humans and a GIO mouse model.

### Yap regulates adult skeletal muscle mass regardless of metabolic phenotype.

Previously, we showed that Yap is a positive regulator of adult skeletal muscle mass[19]. Considering the differential levels of Yap that we observed in obese human and mouse skeletal muscle groups, we sought to determine if the anabolic effects of Yap were dependent on the intrinsic metabolic properties of a muscle group. Cohorts of 4-week-old C57BL/6J mice were treated with AAV6:Yap-shRNA or AAV6:lacZ-shRNA vectors by intravenous administration to knockdown Yap in muscles throughout the body, including the TA, Gastrocnemius (Gastroc) and Soleus hindlimb muscles which exhibit phenotypically distinct metabolic and fibre type properties[26]. Body composition analysis conducted over 8 weeks following treatment demonstrated that mice expressing Yap-shRNA exhibited a reduced accumulation of body mass and lean mass compared with mice administered AAV6:lacZ-shRNA (Supplementary Fig. 3A, B). No difference was observed in total fat mass or average daily food intake between groups (Supplementary Fig. 3C, D). Western blot analysis performed on lysates of all muscle groups excised at experimental endpoint confirmed knockdown of Yap protein in muscles following treatment, but no difference in Yap protein abundance in the livers of treated mice (Supplementary Fig. 3E–H). Consistent with previous reports in the TA muscle of adult mice[19], Yap knockdown elicited atrophy in the TA ($-11 \pm 3\%$), Gastroc ($-17 \pm 3\%$), and Soleus ($-22 \pm 5\%$ of mean values for mice administered control lacZ-shRNA vector) muscles (Supplementary Fig. 4A). In contrast, no evidence of atrophy was observed in the heart (Supplementary Fig. 4A). The reduction in mass observed in all limb muscles assessed following Yap knockdown was due to a diminished muscle fibre cross-sectional area (Supplementary Fig. 4B). No evidence of muscle fibre degeneration or centrally located nuclei (indicative of ongoing muscle regeneration) was observed on histological

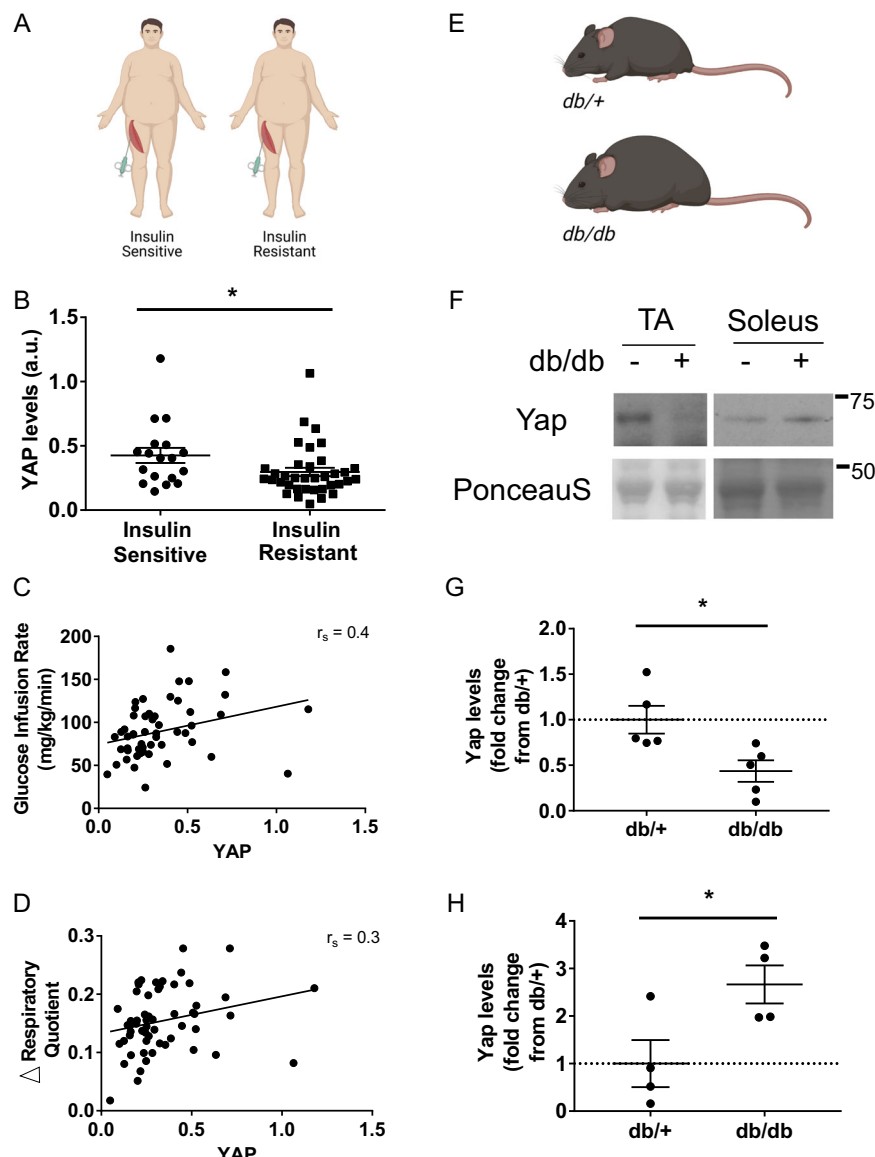

**Fig. 1 YAP levels are reduced in insulin-resistant human and mouse skeletal muscles. A** Muscle biopsies from the vastus lateralis were obtained from obese individuals who were identified as insulin-sensitive or insulin-resistant in skeletal muscle, (**B**) total YAP levels in obese, insulin-sensitive and resistant humans ($n = 54$ total; 18 insulin sensitive and 36 insulin resistant, mean ± SEM, * shows significant difference, $p$ value = 0.04, two-sided $t$ test), (**C**) YAP levels correlated to glucose infusion rate relative to fat free mass (GIR/FFM; $n = 54$, $r_s = 0.4$, $p = 0.02$, and Spearman's rank correlation test), (**D**) YAP levels correlated to difference in respiratory quotient between fasting to glucose-infused state (fed ΔRQ) ($n = 54$, $r_s = 0.3$, $p = 0.016$, and Spearman's rank correlation test), (**E**) Yap levels were assessed in the tibialis anterior (TA) and soleus muscles of lean (db/+) and obese (db/db) mice at 6 weeks of age, (**F**) representative western blots of Yap and Ponceau S levels in db/+ (−) and db/db (+) TA and soleus muscles at 6 weeks of age ($n = 5$ and 4 respectively), (**G**) quantification of total Yap levels in TA samples shown in (**A**) ($n = 5$, mean ± SEM, * shows significant difference, $p$ value = 0.01, two-sided $t$ test), and (**H**) quantification of total Yap levels in soleus samples shown in (**A**) ($n = 4$, mean ± SEM, * shows significant difference, $p$ value = 0.01, two-sided $t$ test).

sections in any muscle group (Supplementary Fig. 4C). Together, these findings demonstrate that reductions in the levels of Yap in adult skeletal muscle fibres impacts muscle fibre size independently of the muscles' intrinsic metabolic phenotype.

**Inhibition of Yap impairs fatty acid oxidation and leads to lipotoxicity in adult skeletal muscle.** To explore the functional consequences of inhibiting Yap on the metabolic properties of skeletal muscle, we first performed steady state-polar metabolomic analysis of Soleus muscles collected 28 days after treatment with AAV6:Yap-shRNA or AAV6:lacZ-shRNA vectors. In total, we detected 297 metabolite profiles (Supplementary Table 1).

Hierarchical clustering revealed robust changes in metabolites between conditions (Supplementary Fig. 5A). Metabolites that were decreased in response to Yap knockdown included fatty acids (Undecanoic acid, Capric acid, 2-Octenoic acid) and 2-oxoglutaric acid (alpha-ketoglutarate), a key metabolite of the TCA cycle, suggesting impaired fatty acid metabolism occurs in adult skeletal muscles with reduced Yap abundance (Fig. 2a, Supplementary Fig. 5A, Supplementary Table 1). Soleus muscles administered AAV6:Yap-shRNA also displayed reduced levels of a number of amino acids (L-Lysine, L-Serine, L-Proline and Aspartic Acid), Ureidopropionic acid and Carnosine, indicating alterations in amino acid, uracil and carnosine metabolism pathways (Fig. 2a, Supplementary Fig. 5A, Supplementary

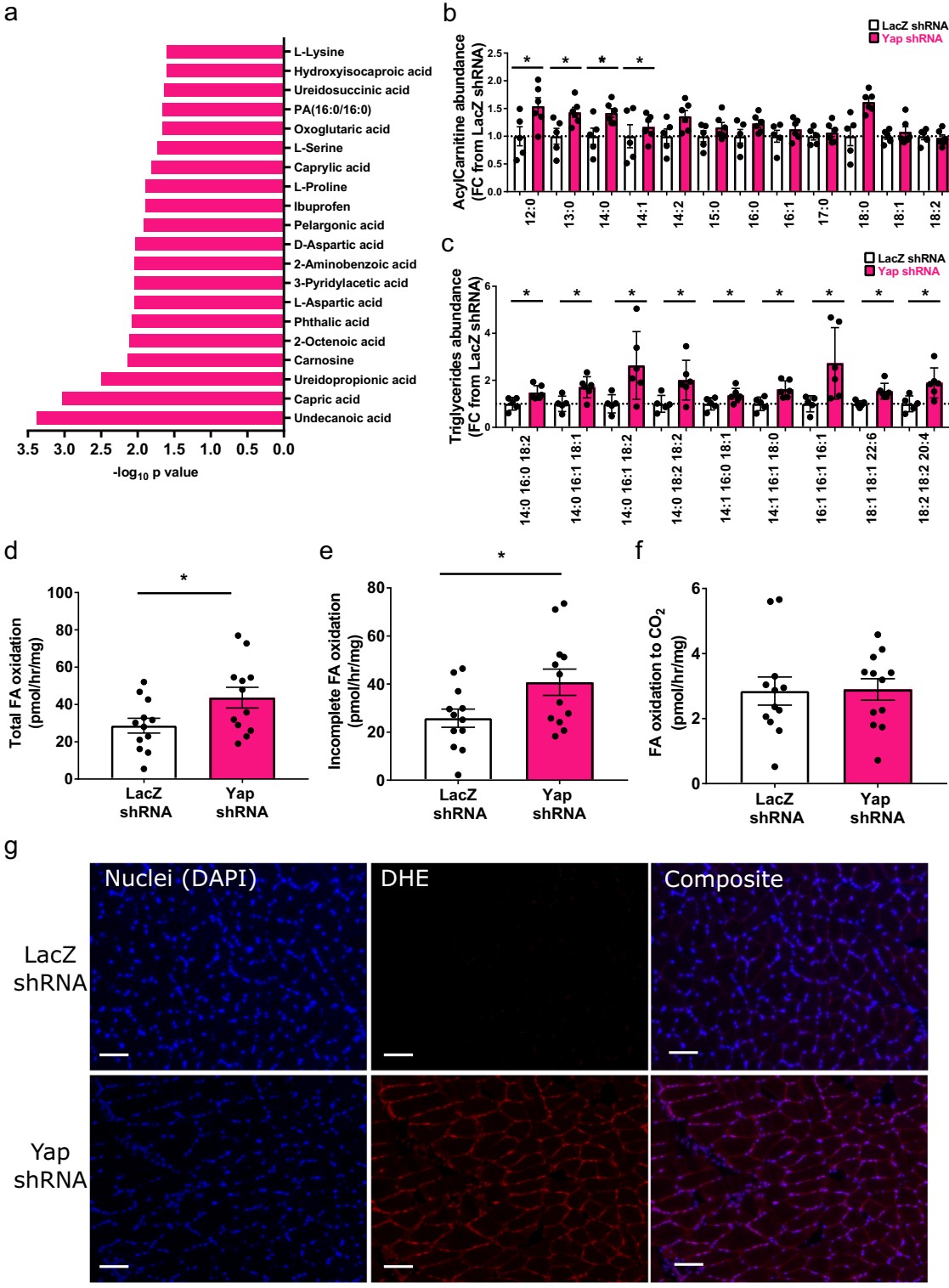

Table 1). Based on these observations, we profiled changes to the skeletal muscle lipidome. Muscles treated with AAV6:Yap-shRNA accumulated increased quantities of long-chain acylcarnitines and select triglyceride species demonstrating that sustained reductions in Yap in adult skeletal muscle leads to an accumulation of neutral lipids and fatty acid derived-intermediates (Fig. 2b, c, Supplementary Fig. 5B, Supplementary Table 1).

To establish the functional significance of Yap in the context of lipid metabolism, we submitted intact EDL muscles of 8-week-old

C57BL/6J male mice treated with AAV6:Yap-shRNA or AAV6: lacZ-shRNA to ex vivo fatty acid oxidation assays using $^{14}C$ labelled oleate. Using this approach, we were able to assess the effect of modulating Yap abundance on fatty acid metabolism specifically in skeletal muscles, thus mitigating potential secondary effects from other organs. The knockdown of Yap in EDL muscles increased total fatty acid oxidation rates, which were determined to be a product of higher rates of incomplete oxidation to β-oxidation intermediates, with no difference in the rates of complete oxidation to $CO_2$ (Fig. 2d–f). No differences

**Fig. 2 Inhibition of Yap impairs fatty acid oxidation and leads to lipotoxicity in adult skeletal muscle. a** Top 20 downregulated metabolites in muscles treated with AAV6:Yap shRNA ranked by $-\log_{10} p$ value, (**b**) Acylcarnitine levels in Soleus muscles examined 4 weeks after administration of AAV6:LacZ shRNA or AAV6:Yap shRNA vector ($n = 6$, mean ± SEM, * indicates $p < 0.05$ with exact $p$ values provided in source data, two-sided $t$ test without adjustment for multiple comparisons), (**c**) selected triglyceride species in Soleus muscles treated as in (**b**) ($n = 6$ biologically independent animals, mean ± SEM, * indicates $p < 0.05$ with exact $p$ values provided in source data, two-sided $t$ test without adjustment for multiple comparisions), (**d**) Total fatty acid oxidation (tracer present in cell and media acid-soluble metabolites and $CO_2$ gas phase) in pmol/h/mg tissue ($n = 12$ biologically independent animals, mean ± SEM, * shows significant difference, $p = 0.03$, two-sided $t$ test), (**e**) Incomplete fatty acid oxidation (tracer in cell and media acid-soluble metabolites) and (**f**) complete fatty acid oxidation to $CO_2$ (tracer present in $CO_2$ gas phase only) in pmol/h/mg tissue in EDL muscles analysed 2 weeks after injection of AAV6:lacZ-shRNA or AAV6:Yap-shRNA vectors ($n = 12$ biologically independent animals, mean ± SEM, * shows significant difference, $p = 0.03$ and 0.71 respectively, two-sided $t$ test), (**g**) Representative images of Dihydroethidium (red) and DAPI (blue) staining in Soleus muscles treated with AAV6:lacZ-shRNA and AAV6:Yap-shRNA and assessed 8 weeks after treatment ($n = 10$ biologically independent animals, scale bars indicate 50 μm).

were observed to the rates of fatty acid uptake or triglyceride esterification (Supplementary Fig. 6A, B). Incomplete fatty acid oxidation and the accumulation of lipid intermediates has been reported to lead to lipotoxicity and is associated with DNA fragmentation and increased reactive oxygen species (ROS)[4,27-34]. Consistent with these reports, the sustained inhibition of Yap in adult skeletal muscle fibres led to a robust increase in the numbers of muscle fibre nuclei with fragmented DNA and ROS, as determined by the presence of TUNEL and Dihydroethidium reactivity respectively (Fig. 2g, Supplementary Fig. 6C).

Skeletal muscle from obese individuals is characterised by perturbed mitochondrial function due to a disconnect between the rates of beta-oxidation and the capacity of the TCA cycle, rather than defects in the electron transport chain directly[4,29,35]. To confirm that the knockdown of Yap in adult limb muscles did not impact mitochondria function directly, we isolated mitochondria from Gastroc muscles 8 weeks after administration of AAV6:Yap-shRNA or AAV6:lacZ-shRNA treatments to measure mitochondrial oxygen consumption rates. No differences were observed in the rates of basal or stimulated oxygen consumption in isolated mitochondria from muscles treated with AAV6:Yap-shRNA or AAV6:lacZ-shRNA vectors (Supplementary Fig. 6D). To validate these findings by an independent methodology, we also assessed respiratory capacity in intact EDL limb muscles where mitochondria remain within their biological niche. Consistent with our findings in isolated mitochondria, no difference was observed in mitochondrial respiratory capacity between intact EDL muscles administered AAV6:Yap-shRNA or AAV6:lacZ-shRNA suggesting that Yap knockdown is unlikely to directly influence electron transport function or respiratory capacity in mitochondria (Supplementary Fig. 6E). Together, these findings support the conclusion that Yap is required for efficient fatty acid oxidation in adult skeletal muscle fibres, such that loss of Yap in skeletal muscle results in presentation of critical features of lipotoxicity and metabolic disease.

**Yap regulates a transcriptional programme that impacts skeletal muscle metabolic substrate utilisation.** To gain insight into the molecular changes that occur following Yap knockdown in adult skeletal muscle fibres, we administered AAV6:Yap-shRNA and AAV6:lacZ-shRNA vectors to the contralateral Gastroc and TA muscles of 8-week-old male C57BL/6J mice. At 28 days post-treatment, muscle fibre nuclei were isolated by flow cytometry following labelling with the peri-nuclear marker PCM1[36,37] (Fig. 3A). Successful isolation of adult muscle fibre nuclei in the PCM1$^{+ve}$ fraction was demonstrated by enrichment of muscle-specific transcripts encoding for *Myogenin* (*Myog*) and *Myostatin* (*Mstn*) relative to PCM1$^{-ve}$ nuclei (Supplementary Fig. 6A). In contrast, PCM1$^{-ve}$ nuclei were enriched for transcripts typically expressed in fibroblasts, including *Collagen1a1 and 3a1* (*Col1a1*, *Col3a1*) and *Fibronectin* (*Fn*) (Supplementary Fig. 7A). RNA extracted from pooled PCM1$^{+ve}$ nuclei was used to prepare

sequencing libraries for Assay for Transposase-Accessible Chromatin (ATAC) and RNA-sequencing to assess global chromatin accessibility and gene expression respectively[38]. In parallel, the Gastroc muscles of a separate cohort of mice treated in the same manner were isolated for label-free proteomics (Fig. 3A).

ATAC-sequencing in PCM1$^{+ve}$ nuclei from limb muscles mapped a total of 36,097 peaks across samples representing regions of accessible chromatin that were detected in at least two independent samples. Of these, 275 peaks were different between muscles treated with AAV6:lacZ-shRNA and AAV6:Yap-shRNA, with 42 regions more accessible in limb muscles administered AAV6:Yap-shRNA and 233 more accessible in limb muscles administered AAV6:lacZ-shRNA (Fig. 3B, Supplementary Table 2; FDR < 0.05). Regions of chromatin that became less accessible following Yap knockdown were located largely within intergenic regions (30.1 vs. 40.1% of peaks in YAP shRNA and lacZ-shRNA muscles respectively: Supplementary Fig. 7B). In contrast to the reduced accessibility in intergenic regions, chromatin accessibility at promoters (1 kb ± TSS) was increased in limb muscles expressing Yap shRNA compared to muscles expressing lacZ-shRNA (16.3 vs. 10.5% of ATAC-peaks: Supplementary Fig. 7B).

RNA-sequencing from the same pool of isolated muscle nuclei detected 6269 transcripts that were differentially expressed following Yap knockdown, comprising 3851 upregulated and 2418 downregulated genes (Fig. 3C, Supplementary Table 3; FDR < 0.05). To assess the impact of global changes in gene expression, we performed Gene-set enrichment analysis (GSEA) of RNA-sequencing data[39]. GSEA identified an increase in gene expression associated with immune response and inflammatory pathways (Fig. 3D, Supplementary Table 4) and a downregulation in gene expression associated with actin folding by CCT/TriC and the TCA cycle in limb muscles expressing Yap shRNA (Fig. 3E, Supplementary Table 4).

In parallel to our investigation of the effects of depressing Yap activity on the transcriptome, we measured the impact of Yap knockdown on the skeletal muscle proteome. Applying label-free quantitative proteomics to mouse Gastroc muscles treated with AAV6:Yap-shRNA or AAV6:lacZ-shRNA, we quantified 1512 distinct proteins (Fig. 3F, Supplementary Table 5). The abundance of 150 proteins was significantly different following Yap knockdown (Fig. 3F, Supplementary Table 5; adjusted $p$ value < 0.05). In total, 119 proteins were increased in abundance that included metabolic enzymes (Suclg2, Ampd1) and components of the Polyamine signalling pathway (Amd1, Smox), and 31 proteins were reduced in abundance including the enzymes Pdk4 and Acadsb that play key roles in substrate utilisation and lipid metabolism, respectively (Fig. 3F, Supplementary Table 5; adjusted $p$ value < 0.05). GSEA analyses of differentially expressed proteins in limb muscles expressing Yap-shRNA supported our transcriptomics data and revealed an increase in the abundance of proteins associated with the catabolism of carbohydrates and a decrease in the abundance of proteins associated with the

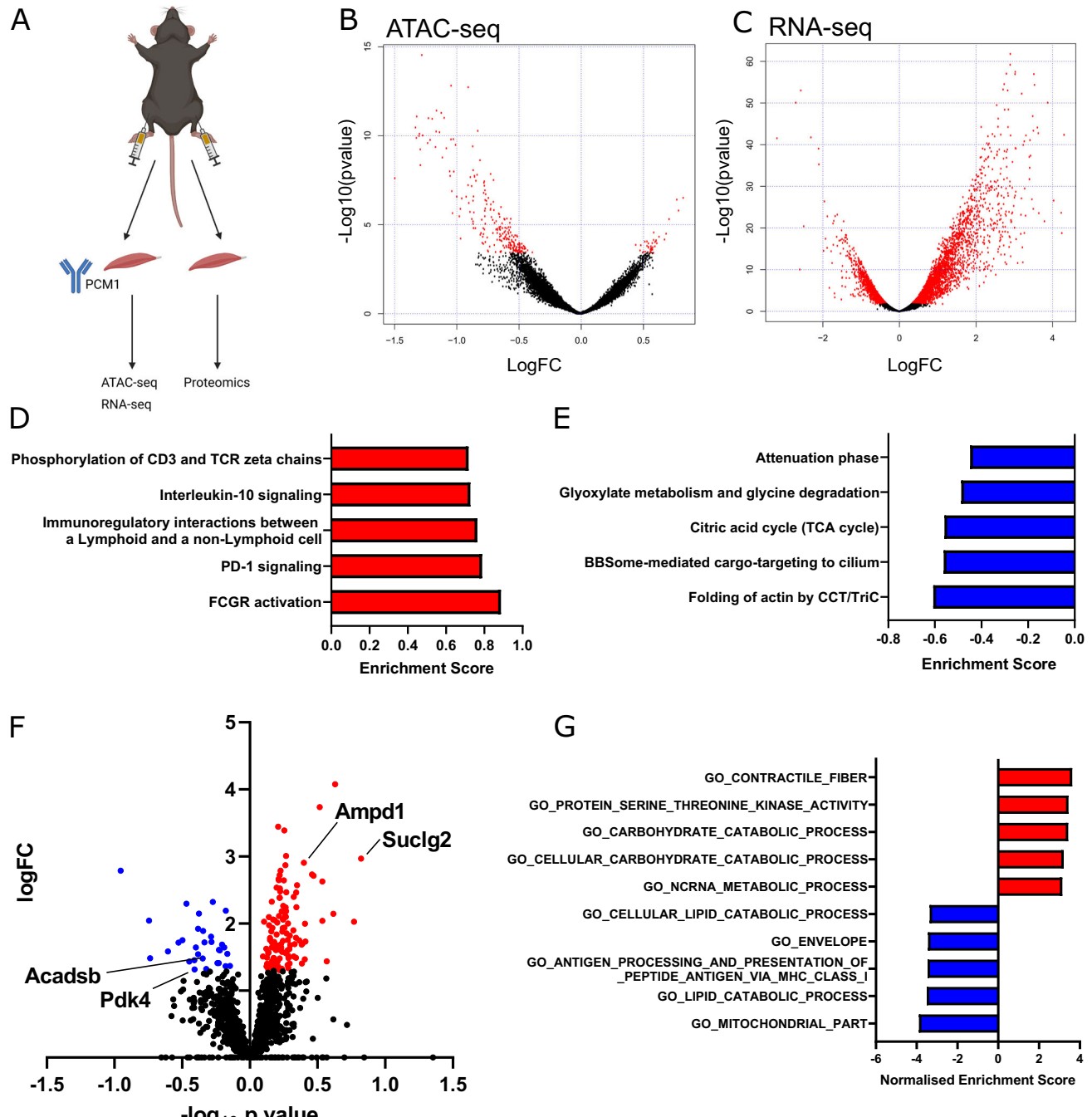

**Fig. 3 Yap influences a transcriptional programme that impacts skeletal muscle metabolic substrate utilisation. A** Schematic demonstrating integrated transcriptomics and proteomics experimental design including PCM1 cell sorting strategy to isolate adult myonuclei from 8-week-old C57BL/6 male mouse limb muscles at 28 days following treatment with AAV6:lacZ-shRNA or AAV6:Yap-shRNA, (**B**) MA plot of ATAC-sequencing showing more and less accessible regions of chromatin following Yap knockdown. Differential peaks are shown in red ($n = 3$ biologically independent animals; adjusted $p$ value < 0.01, multiple comparisons were corrected using Benjamini–Hochberg procedure), (**C**) MA plot of RNA-sequencing showing up and downregulated transcripts following Yap knockdown. Differential transcripts are shown in red ($n = 3$ biologically independent animals; adjusted $p$ value < 0.01, multiple comparisons were corrected using Benjamini–Hochberg procedure), (**D**) top five upregulated and (**E**) top five downregulated GSEA terms from RNA-seq data generated in (**C**), (**F**) volcano plot of proteins detected using label-free proteomics in muscles treated as in (**A**). Grey dots show proteins that were not significantly different, blue dots show proteins that were significantly downregulated and red dots proteins that were significantly upregulated ($n = 5$ biologically independent animals; adjusted $p$ value < 0.01, $t$ test with correction for multiple comparisons using Benjamini–Hochberg procedure), (**G**) GSEA analysis of proteomics datasets. Top five upregulated terms are shown in red, top five downregulated terms in blue.

catabolism of fatty acids and cellular lipids (Fig. 3G, Supplementary Table 6). Together, these findings demonstrate that Yap regulates a transcriptional programme that governs metabolic substrate utilisation in adult skeletal muscle.

**Idh2 upregulation prevents impaired fatty acid oxidation associated with Yap knockdown.** To further explore how the observed differences in gene expression impact fatty acid metabolism in adult muscles expressing Yap-shRNA, we mapped

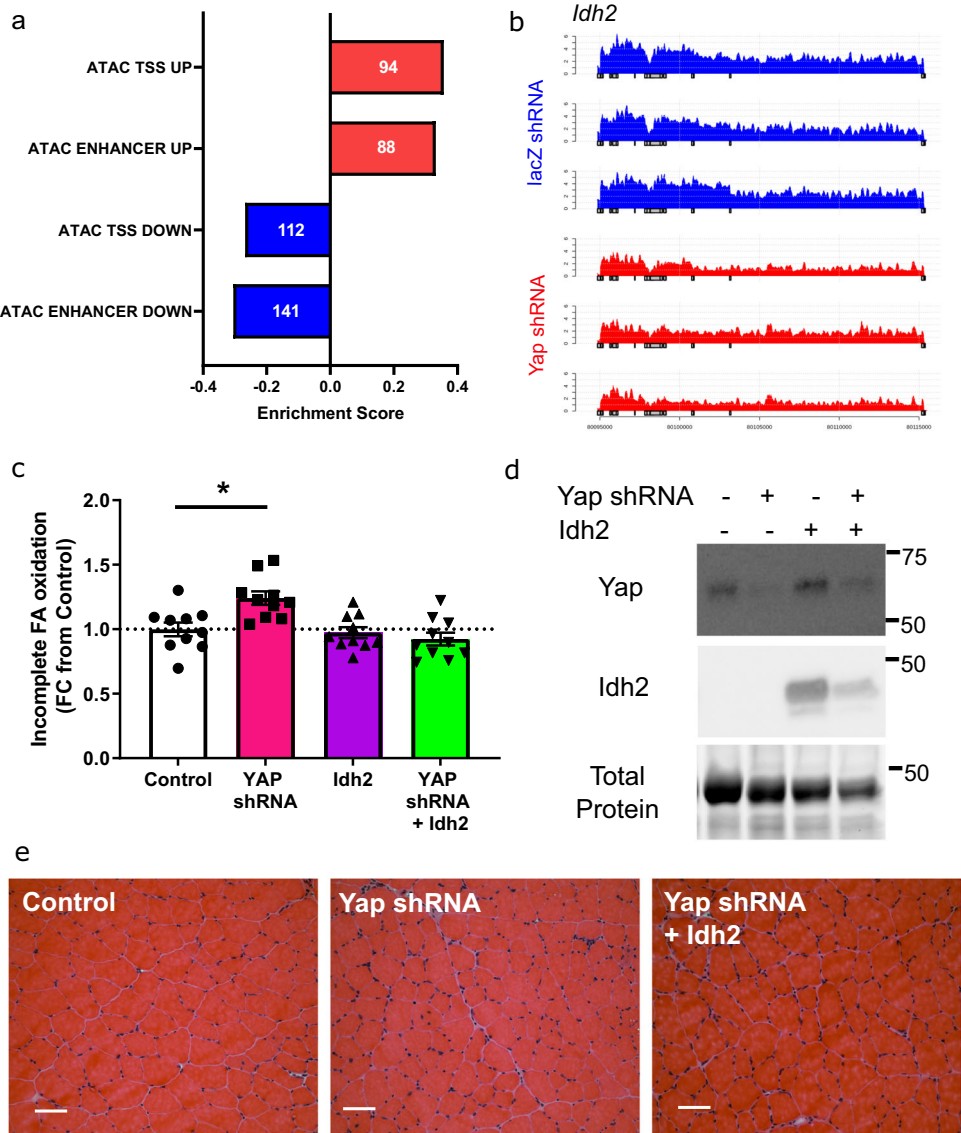

**Fig. 4 Idh2 upregulation prevents impaired fatty acid oxidation associated with Yap knockdown. a** Graph showing the changes in chromatin accessibility and transcript abundance at enhancers or promoters (TSS). Numbers within bars indicate the number of genes detected in each comparison (FDR < 0.05, rank-MANOVA), (**b**) sequencing tracks of *Idh2* transcript in AAV6:lacZ-shRNA (blue) and AAV:Yap shRNA (red). Each track represent a biological replicate from an individual animal (*n* = 3), (**c**) incomplete fatty acid oxidation in EDL muscles analysed 2 weeks after injection of control, AAV6:Yap-shRNA, AAV6:Idh2 vectors or both (*n* = 10 biologically independent animals, mean ± SEM, * shows significant difference, one-way ANOVA with Tukey's multiple comparison test with exact *p* values provided in source data), (**d**) representative western blots of Yap, Idh2, and total protein levels in TA muscles from mice treated as in (**c**) (*n* = 5 biologically independent animals), (**e**) Representative H and E images of TA muscle from mice treated as in (**c**) (*n* = 5 biologically independent animals, scale bars indicate 50 μm).

changes in chromatin accessibility identified by ATAC-sequencing at enhancers or promoters to target gene transcript abundance. Using this approach, we identified 141 regions of chromatin identified as enhancers that became less accessible and were associated with a reduction in target gene expression, and 88 enhancers that were more accessible with an increase in target gene expression following Yap knockdown (Fig. 4a, Supplementary Table 3, FDR < 0.05). A total of 112 promoters (1 kb ± transcriptional start site) were less accessible, and 94 promoters more accessible, with a corresponding directional change in target gene expression in Yap-shRNA expressing muscle nuclei (Fig. 4a, Supplementary Table 3). YAP predominately binds the TEAD transcription factors to interact with the transcriptome at distal enhancers[14–16]. Consequently, we focussed our analysis on the 141 genes that were downregulated and associated with enhancers

that displayed reduced accessibility following Yap knockdown. The top downregulated genes included established mediators of substrate metabolism and the TCA cycle (Slc2a3/Glut3, Gpt2, Idh2 and Glul), glucocorticoid signalling (Nr3c1, Sgk1) and G-Protein coupled receptor signalling (Adrb2, Arrdc2 and Prkce) (Supplementary Table 3). Of these genes, 28 were associated with enhancers that contain a TEAD binding motif including the enzyme, Idh2 (Fig. 4b, Supplementary Table 3).

IDH2 is a mitochondrial localised enzyme within the TCA cycle that catalyses the oxidative decarboxylation of isocitrate to alpha-ketoglutarate and a reduction in NADP to NADPH$^+$ levels[40,41]. Consequently, IDH2 is required for efficient energy substrate metabolism and redox signalling[40]. Considering our observations that muscles with reduced Yap levels display impaired fatty acid oxidation, features of lipotoxicity and lower

alpha-ketoglutarate levels (Fig. 2), we sought to test if over-expression of Idh2 could prevent aspects of the metabolic perturbations observed in muscles following Yap knockdown. Cohorts of 8-week-old C57BL/6J male mice were treated in TA/EDL muscles by direct intramuscular injection with AAV6:Yap shRNA, AAV6:Idh2 or both vectors in combination. After 14 days, muscles were excised and EDL muscles were subjected to ex vivo $^{14}$C-oleate tracers assays as described in Fig. 2. While knockdown of Yap resulted in incomplete fatty acid oxidation, this effect was completely prevented by co-expression of Idh2 (Fig. 4c). Idh2 expression alone had no impact on fatty acid uptake, triglyceride esterification, or fatty acid oxidation rates (Fig. 4c, Supplementary Fig. 8A, B). Yap knockdown and Idh2 expression were confirmed in TA muscles by Western blotting (Fig. 4d, Supplementary Fig. 8C). Histological analysis of muscles revealed no detrimental impact of any intervention (Fig. 4e). Together, these findings show that Yap impacts skeletal muscle metabolism and substrate utilisation by transcriptional control of metabolic-related genes including Idh2.

**Increasing Yap in the musculature of obesogenic *db/db* mice enhances energy expenditure to reduce adiposity and hepatic steatosis.** In light of the reduced levels of YAP in the muscles of obese insulin resistant humans and mice, and the deleterious metabolic effects of experimentally reducing Yap protein in mouse muscles, we sought to test the functional importance of skeletal muscle Yap during the progression of metabolic disease. A single systemic administration of AAV6 vectors encoding mouse Yap (AAV6:Yap), or AAV6:empty vector was delivered via tail vein injection to 4-week-old *db/db* mice to achieve body-wide transduction of skeletal musculature[42]. Following treatment, we observed that *db/db* mice administered AAV6:empty vector continued to gain body mass throughout the study duration, whereas from as early as 4 weeks post-treatment, *db/db* mice treated with AAV6:Yap failed to accumulate body mass at the same rate as mice administered AAV6:empty vector (Fig. 5a, b). The observed differences in body mass as a consequence of treatment with AAV6:Yap were driven exclusively by a failure to accumulate additional fat mass (Fig. 5c), as no differences in lean mass were observed between *db/db* mice treated with AAV6:Yap and AAV6:empty vector (Supplementary Fig. 9A). Expression of exogenous Yap in the Gastroc hindlimb muscles of *db/db* mice administered AAV6:Yap was confirmed by Western blotting (Supplementary Fig. 9B). No increase in Yap levels was observed in the livers of treated mice, demonstrating specific transduction of the striated musculature following treatment (Supplementary Fig. 9C). Of note, Yap mRNA expression in treated mice correlated with the expression of established target genes Cyr61, Ctgf and Amotl2, as well as the expression of Idh2, in line with our findings in non-obese C57BL/6J mice (Supplementary Fig. 10A–D). No difference in the mass of limb muscles, or the heart was detectable at 14 weeks after vector administration (Supplementary Fig. 10E–G). Average daily food intake for *db/db* mice treated with AAV6:Yap was similar between *db/db* mice administered AAV6:empty vector indicating that the restoration of Yap in skeletal musculature limited the accumulation of adiposity independent of changes in lean mass or food intake, and in spite of continued excess energy intake (Fig. 5d).

While persistently hyperphagic *db/db* mice that overexpressed Yap in striated muscles continued to exhibit fasting hypergly-caemia and impaired whole-body glucose tolerance (Supplementary Fig. 11A, B), fasting plasma concentrations of insulin (Fig. 5e) and c-peptide levels (Supplementary Fig. 11C) were lower indicative of a reduction in insulin secretion. To ascertain if the upregulation of Yap in skeletal muscles improved glucose-

mediated insulin secretion, we administered a physiologically relevant bolus of glucose via oral gavage to *db/db* mice treated with AAV6:Yap or AAV6:empty vector and assessed plasma insulin secretion. Following glucose challenge, plasma insulin concentrations increased in *db/db* mice previously treated with AAV6:Yap, but not in *db/db* mice administered AAV6:empty vector (Fig. 5f), demonstrating that glucose-stimulated insulin secretion was enhanced in *db/db* mice expressing increased levels of Yap protein in the striated musculature.

To gain further insight as to how Yap over-expression in the skeletal muscles of *db/db* mice limited adiposity, we performed metabolic caging studies to determine whole-body metabolic function. While no significant differences were observed in total activity in active or inactive phases (Supplementary Fig. 11D, E), *db/db* mice treated with AAV6:Yap displayed a marked increase in energy expenditure at the onset of the active phase that was independent of total lean mass (Fig. 5g, h). Histological analysis of peripheral tissues 14 weeks after systemic administration of AAV6:Yap to *db/db* mice confirmed a reduction in adipocyte size, in line with body composition data (Fig. 5i). Strikingly, we found that inguinal adipose tissues of *db/db* mice treated with AAV6:Yap contained beige adipocytes as indicated by the presence of multilocular adipocytes that exhibited immunoreactivity to the beige adipocyte marker, Ucp-1 (Fig. 5i, j). Elevated Ucp-1 expression in white adipose tissue was confirmed by qPCR (Supplementary Fig. 11F). Importantly, no increase in Yap mRNA expression was detected in adipose tissue further confirming that the expression of AAV6:Yap was restricted to the striated musculature (Supplementary Fig. 11G). Additionally, muscle-directed Yap gene delivery was accompanied by an increased intensity of histochemical reaction for succinate dehydrogenase activity in TA limb muscles (Supplementary Fig. 11H) and protection from the development of liver steatosis (Fig. 5k). Collectively, these findings demonstrate that the observed reduction in Yap protein in the skeletal muscles of obese insulin-resistant *db/db* mice contributes to whole-body and organ-specific alterations in lipid homoeostasis. Activating Yap in striated muscle is associated with an increase in energy expenditure, browning of white adipose tissue and increased skeletal muscle oxidative potential. Together, these effects result in an attenuation of the progression of pathological features of hyperphagia-induced obesity.

## Discussion

While the Hippo signalling pathway has been defined as a reg-ulator of cell fate, apoptosis and proliferation in many tissues, several studies have now demonstrated that the Hippo signalling pathway also regulates cell size in adult skeletal muscle[19,20,43]. Seeking to further define the role of the Hippo pathway in mature skeletal musculature, we report herein a metabolic function for Yap in adult skeletal muscle fibres, such that sustained inhibition of Yap alters metabolic substrate utilisation leading to incomplete fatty acid oxidation.

A key finding of our study is that Yap influences the rate of oxidation of fatty acids in skeletal muscle, where inhibition of Yap results in elevated, but incomplete, fatty acid oxidation. While initially thought to be associated with reduced rates of β-oxida-tion, a growing body of evidence supports the concept that ele-vated rates of incomplete β-oxidation without concurrent increases in Krebs/TCA cycle intermediates or mitochondrial respiratory capacity is a critical event in the development and progression of metabolic stress[4,29,32]. Recent studies have demonstrated causal interactions between fatty acid metabolism and skeletal muscle mass[29,44–46]. However, in obesity, anabolic signalling in skeletal muscle is perturbed[47]. Our observations

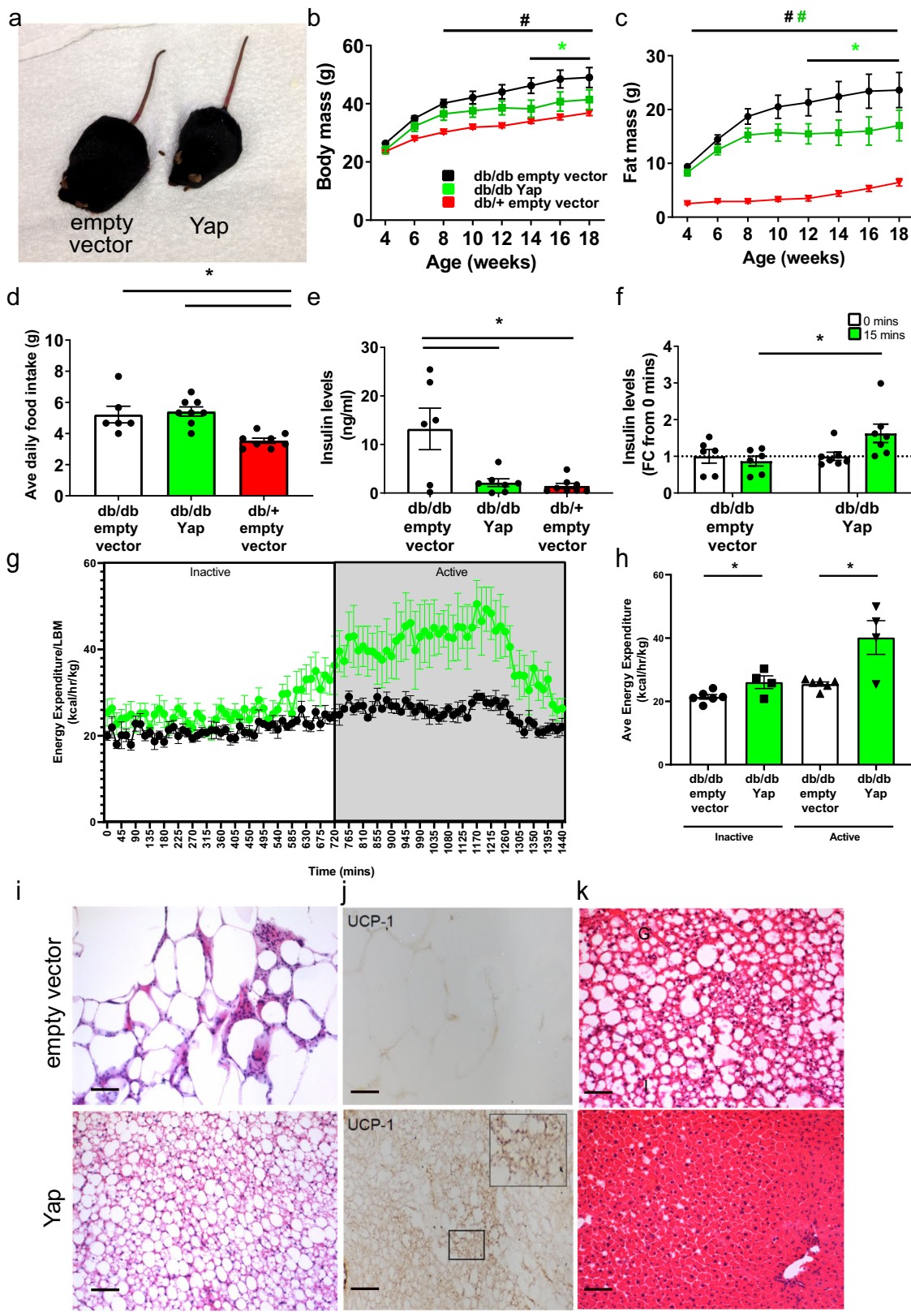

support these conclusions; while Yap is sufficient, and required, for muscle growth in healthy adult mice[19,20], Yap over-expression in obese mice led to improved metabolic function without changes in lean mass. These findings suggest that the metabolic and anabolic consequences of modulating the Hippo pathway in adult skeletal muscle are distinct processes. Further work

exploring the essential genes that regulate the anabolic and metabolic programmes controlled by YAP in skeletal muscle may provide new insight into how the response to anabolic interventions is impaired by obesity. Our study identifies a previously unappreciated link between the Hippo pathway and metabolism in the post-mitotic cell. Our findings demonstrating Yap

**Fig. 5 Increasing Yap in the musculature of obesogenic *db/db* mice enhances energy expenditure to reduce adiposity and hepatic steatosis. a** Representative image of *db/db* mice 10 weeks after administration of AAV6:empty vector or AAV6:Yap vectors (n = 7 *db/db* empty vector, 9 *db/db* Yap, 8 *db/+*) (**b**) body mass and (**c**) fat mass (in grams) of *db/db* mice over 14 weeks following administration of AAV6:empty vector (black) or AAV6:Yap (green) vectors. *db/+* mice treated with AAV6:empty vector included as a control (red) (mean ± SEM, *indicates $p < 0.05$ with exact p values provided in source data, n = 7 *db/db* empty vector, 9 *db/db* Yap, 8 *db/+*, two-way ANOVA with Tukey's multiple comparisons test)), (**d**) average daily food intake (g = grams) of mice treated as in (**b**) after 12 weeks (n = 6 *db/db* empty vector, 8 *db/db* Yap, 8 *db/+*, mean ± SEM, * indicates $p < 0.05$ with exact p values provided in Source Data, one-way ANOVA with Tukey's multiple comparisons test), (**e**) fasting plasma insulin concentrations (ng/ml) of mice treated as in **b**) after 14 weeks of treatment (n = 6 *db/db* empty vector, 7 *db/db* Yap, 8 *db/+*, mean ± SEM, *indicates $p < 0.05$ with exact p values provided in source data, one-way ANOVA with Tukey's multiple comparisons test), (**f**) plasma insulin concentrations in mice treated as in (**b**) after 12 weeks following 6 h fast (0 min) or oral gavage of glucose (15 min) (n = 6 *db/db* empty vector, 7 *db/db* Yap, mean ± SEM, * indicates $p < 0.05$ with exact p values provided in source data, one-way ANOVA with Sidak's multiple comparisons test), (**g–h**) Energy expenditure (kcal/hr/kg lean body mass) of mice treated as in (**b**) after 10 weeks of treatment. 12 h inactive and 12 h active cycles are indicated in (**g**) with quantification shown in (**h**) (n = 6 *db/db* empty vector, 4 *db/db* Yap, mean ± SEM, *t* test comparing empty vector and Yap treatments in inactive ($p = 0.03$) or active phases ($p = 0.009$), (**i**) representative H and E image of adipose tissue, (**j**) Ucp-1 labelling of inguinal adipose tissue. Insert shows higher magnification, (**k**) livers of mice stained with H and E following treatment as in (**b**) (**i-k** n = 6 *db/db* empty vector, 8 *db/db* Yap, scale bars indicate 50 μm). All n values biologically independent animals.

functions as a mediator of substrate utilisation and lipid metabolism in adult skeletal muscle contrast with those conducted in mitotically-active epithelial cells where the Hippo pathway is responsive to, and a mediator of, glycolysis and glucose metabolism[48–53]. The differences between our findings and these studies likely reflect the substrate preferences and metabolic flexibility of the different cells/tissues studied, as many groups have examined Hippo pathway effects in cancer cell lines, where metabolic reprogramming towards anaerobic glycolysis is an established feature[54]. While our findings demonstrate that sustained inhibition of Yap in adult skeletal muscle leads to negative metabolic consequences, our observations are also consistent with a model where transient inhibition of Yap may be a necessary feature of a concerted programme of adaptive events to partition substrate utilisation and tune anabolic/catabolic activity during episodes of energy stress. Assessing Yap activity during conditions of energy stress such as caloric restriction or intense bouts of exercise may provide further insight into the physiological role of the Hippo pathway in adult skeletal muscle fibres. While the beneficial metabolic effects of YAP over-expression in obese mice are likely due to the reported increases in energy expenditure, the precise mechanisms for how YAP activation impacts energy expenditure and leads to browning of adipose tissue remain elusive. Future studies exploring skeletal muscle intrinsic and extrinsic factors known to mediate tissue cross-talk in obesity such as myokines are required[55].

In conclusion, we demonstrate that, in addition to the anabolic function of Yap in healthy adult skeletal muscle, the Hippo pathway regulates the expression of genes required for efficient substrate utilisation and fatty acid oxidation. Our studies show that the Hippo pathway is perturbed in the skeletal muscle of humans and mice with metabolic disease. Importantly, we demonstrate that increasing skeletal muscle Yap activity in pre-clinical models of metabolic disease limits adiposity and hepatic steatosis by increasing energy expenditure. Our findings identify the Hippo pathway effector Yap as a regulator of skeletal muscle attributes with consequences on whole-body and organ-specific lipid homoeostasis. Interventions that target Yap may offer opportunities to enhance skeletal muscle lipid metabolism with implications for the treatment of metabolic disease.

## Methods
### Ethical approval.
Animal experiments were conducted in accordance with the relevant codes of practice for the care and use of animals for scientific purposes as stipulated by the National Health and Medical Research Council of Australia and conducted with approval from the respective Animal Ethics Committees of the Alfred Medical Research and Education Precinct (AMREP, E/1659/2015/B), and the University of Melbourne (Melbourne, Australia, 1714353 and 1814680). Human muscle biopsies were procured previously with ethical consent provided by the St Vincent's Hospital Human Research Ethics Committee (HREC/10/SVH/133. Sydney, Australia). All relevant ethical regulations were complied with and informed consent was provided by study participants.

### Reagents.
Reagents were purchased from Sigma-Aldrich, Australia unless stated otherwise.

### Generation of serotype 6 recombinant adeno-associated viral (AAV) vectors.
DNA sequences that encode mouse Yap or Yap-shRNA were synthesised by Genscript and sub-cloned into AAV6:CMV-MCS-SV40pA or AAV6:CMV-eGFP-MCS-SV40pA plasmids respectively. DNA sequences that encode mouse Idh2 were synthesised by Vectorbuilder and sub-cloned into AAV6:CMV-MCS-SV40pA. Empty vector (AAV6:CMV-MCS-SV40pA; called AAV6:empty vector), or shRNA targeting lacZ (AAV6:lacZ-shRNA) were used as controls. Recombinant adeno-associated viral vectors were generated by co-transfection of 10 μg of plasmids containing cDNA constructs with 20 μg of pDGM6 packaging plasmid into HEK293 cells (seeded 16 h prior at a density of $3.2–3.8 \times 10^6$ cells) using the calcium phosphate precipitate method to generate type-6 pseudotyped viral vectors. After 72 h, cells and culture medium were collected and homogenised, before clarification using a 0.22 μm filter (EMD Millipore). Vectors were purified by affinity chromatography using a HiTrap heparin affinity column (GE Healthcare), ultra-centrifuged overnight and resuspended in sterile Ringer's solution. Vector concentration was determined using a customised quantitative PCR reaction (Applied Biosystems).

### Intramuscular and intravenous injections of AAV6 vectors.
Intramuscular injections of AAV6 vectors were performed in male C57BL/6J mice at 8–10 weeks of age. Intravenous injections of AAV6 vectors (for systemic delivery of AAV vectors to striated muscles body-wide) were performed via tail vein in male C57BL/6J or B6.BKS(D)-Lepr^db/J (*db/db*) mice at 4 weeks of age. C57BL/6J mice were bred at AMREP animal services, Melbourne, Victoria. All *db/db* mice were obtained from Jackson laboratories, USA (stock number 000697). All mice were maintained in a 12:12 h light:dark cycle at a temperature of 22 ± 1 °C, a humidity of 40–70%, and provided *ad libitum* access to food and water during the study duration. For intramuscular injection studies, mice were placed under general anaesthesia (2% isoflurane in $O_2$), then administered a subcutaneous injection of carprofen (5 mg/kg) for post-operative analgesia, followed by hindlimb muscle injections of AAV6 vectors diluted in 30 μl of Hank's buffered salt solution (HBSS) to a dosage of $1 \times 10^{10}$ vectors genomes. For systemic muscle transduction studies, conscious mice were administered a tail vein injection of AAV6 vectors diluted in in 100 μl of HBSS to a dosage of $1 \times 10^{12}$ vg. At experimental endpoints, C57BL/6J mice were killed by cervical dislocation in the fed state (within 3 h of the initiation of the light phase) unless indicated. All *db/db* mice were fasted for 5 h commencing within 3 h of the initiation of the light phase. Tissue samples were rapidly excised and snap-frozen for biochemical assessment or processed for histology as described below.

### PCM1 cell sorting of skeletal muscle nuclei for ATAC/RNA-sequencing.
Skeletal muscles were pooled to obtain enough material for sequencing (4 × Gastrocnemius and 4 × tibialis anterior muscles per replicate: n = 3 per condition). The samples were homogenised in 15 mL lysis buffer (320 mmol/L sucrose, 10 mmol/L Tris-HCl (pH = 8), 5 mmol/L CaCl$_2$, 5 mmol/L magnesium acetate, 2 mmol/L EDTA, 0.5 mmol/L EGTA, 1 mmol/L DTT and 1× complete protease inhibitor, all chemical reagents from Sigma-Aldrich) using an electrical homogeniser (T10 basic homogeniser workcentre, IKA 4095500). The lysate was combined with another 15 mL lysis buffer and subsequently homogenised using dounce tissue grinder (Wheaton, #357546, Millville, NJ, USA) for 10–15 strokes. The cell lysate was sequentially filtered through 100, 70, and 40 μm cell strainers (Becton Dickinson,

Franklin Lakes, NJ, 10 USA) and centrifuged to pellet nuclei at $1000 \times g$ for 5 min. Nuclear pellets were resuspended in 4 mL sucrose buffer (1 mol/L sucrose, 10 mmol/L Tris-HCl (pH = 8), 5 mmol/L magnesium acetate, 1 mmol/L DTT, and $1 \times$ complete protease inhibitor). The nuclei suspension was cushioned on top of a $2 \times$ volume of the sucrose buffer, which was centrifuged at $1000 \times g$ for 5 min to pellet nuclei. The nuclei pellets were washed once in 3 mL of nuclei storage buffer (NSB, 440 mmol/L sucrose, 10 mmol/L Tris-HCl (pH = 7.2), 70 mmol/L KCl, 10 mmol/L $MgCl_2$, 1.5 mmol/L spermine and $1 \times$ complete protease inhibitor). The washed nuclei pellets were resuspended in NSB and labelled with myocyte nuclear specific marker, PCM1 (1:200, HPA023374, Sigma-Aldrich) at 4 °C for 45 min. Omitting the primary antibody served as a control for positive labelling. Nuclei were washed twice in NSB and labelled with secondary antibody conjugated with Alexa Fluor 633 (1:500, Thermo Fisher Scientific (Molecular Probes)) at 4 °C for 30 min. Nuclei were washed twice in NSB and resuspended in PBS before being processed for cell sorting to obtain PCM1 labelled (+ve) and unlabelled (−ve) fractions (via FACS ARIA cell sorter, BD Biosciences). Following sorting, nuclei were pelleted at $1500 \times g$ for 20 min.

**Quantitative real-time reverse transcriptase PCR.** Total RNA was extracted from 10 to 20 mg of tissue using TRIzol (Life Technologies) or from isolated nuclei using TRIzol followed by Direct-zolTM RNA MicroPrep kit (R2060, Zymo Research) as per manufacturer's protocols. Reverse transcription was performed on 1 μg of total RNA using Multiscribe reverse transcriptase (Life Technologies) as per manufacturer's protocols. Transcripts were measured using Taqman® fluorogenic primer probe sets (Life Technologies, Victoria, Australia) or Syber green (ThermoScientific). Primer probe set IDs for Taqman assays and primer sequences for Syber Green assays are listed in Supplementary Table 7. Relative differences in transcript abundance were determined using the delta delta Ct method[56].

**ATAC-sequencing and RNA-sequencing.** ATAC-sequencing transposition reaction and library preparation was performed on 78,000 PCM1$^{+ve}$ sorted myonuclei using published protocols[57]. Libraries were sequenced for 150 cycles, 120 million reads, paired-end sequencing on a Novaseq 6000 (Illumina) at the Victorian Clinical Genetics Service (VCGS, Melbourne, Australia). For RNA-sequencing from sorted nuclei, ribosomal RNA was depleted from total RNA using NEBNext rRNA Depletion kit (E6310, New England Biolabs) before generation of sequencing libraries using NEBNext Ultra II Directional Library Prep Kit for Illumina (E7760, New England Biolabs). Library fragment sizes were selected at 200–400 bp in length according to manufacturer's protocol. Libraries were sequenced for 150 cycles, 40 million reads, paired-end-sequencing on a Novaseq 6000 (Illumina) at the VCGS, Melbourne, Australia.

**Bioinformatics.** The reference mouse genome was downloaded from Ensembl (Mus musculus.GRCm38 dna.primaryassembly.fa) alongside Ensembl v96 annotation in GTF format[58]. RNA-seq reads underwent quality trimming using Skewer (Skewer 0.2.2)[59] with minimum base quality threshold of 20 and minimum read length of 20 nt. As a quality control measure, we quantified ribosomal RNA in the sequence dataset. Mouse rRNA sequences were obtained from RefSeq for 5S, 5.8S, 18S and 28S rRNA. A sample of 1 million reads was mapped to the rRNAs with BWAmem. Separately, reads were mapped to the reference genome using STAR aligner (STAR 2.5.3a) for RNA-seq and BWA (BWA 0.7.17) for ATAC-seq with default settings[60,61]. Duplicated reads were removed from the ATAC-sequencing experiment using Samtools (Samtools 1.8)[62]. Peak calling during ATAC-sequencing was performed using MACS 1.4.2-1[63]. Peaks that were detected in at least 2 samples were used to curate a peak set with individual sample's peaks counted to the curated peak sets using FeatureCounts (v1.5.0)[64]. FeatureCounts was used to quantify reads mapped to whole gene with mapping quality ≥10 and otherwise default settings. Genes with fewer than 10 reads per sample on average were discarded prior to statistical analysis. General linear model feature of EdgeR (v3.16.5 and v3.26.8) was used to compare data[65]. Pathway analysis was performed using GSEA software to contrast data to MSigDB and/or Reactome gene sets after ranking by differential abundance score [sign of fold change divided by $\log_{10}$ ($p$ value)][39]. Genes and pathways with FDR < 0.05 were considered significant. To understand Yap directed enhancer regulation, we annotated ATAC-seq regions of interest to their nearby gene regulatory features; TSS and enhancer target gene[66]. Genes with altered ATAC-seq signal ($p$ value < 0.01) were collected to separate gene sets for enhancer/TSS and up/down directions. These gene sets underwent enrichment analysis with mitch using RNA profile data[67]. From the list of genes with reduced RNA expression and enhancer accessibility, we screened for those containing a YAP/TEAD4 binding site in the enhancer (Jaspar motif MA0809) with a BedTools (v2.28.0) based shell script[68]. To create sequence coverage plots, the BedTools genomecov tool was used to convert bam files to bedgraph format. Bedgraph data were imported into R v4.0.3 with rtracklayer (v1.50.0), normalised by library size and plotted with a custom R script.

**Sample preparation for proteomic analysis.** Frozen tissue was tip-probe sonicated in 6 M guanidine chloride containing 10 mM tris(2-carboxyethyl)phosphine and 40 mM chloroacetamide in 100 mM Tris pH 8.5. The lysate was incubated at 95 °C for 5 min and centrifuged at $20,000 \times g$ for 30 min at 4 °C. Lysates were

precipitated with 5 volumes of acetone overnight at −20 °C. Protein pellets were centrifuged at $5000 \times g$, 10 min at 4 °C and washed with 80% acetone. Protein pellets were resuspended in 10% trifluoroethanol in 100 mM HEPEs pH 7.5, quantified with BCA and normalised to 20 μg/10 μl. The lysates were digested with 0.4 μg of Sequencing Grade Trypsin (Sigma) overnight at 37 °C. Peptides were acidified with 100 μl of 1% trifluoroacetic acid (TFA) and purified using styrenedivinylbenzene- reverse phase sulfonate microcolumns. The columns were washed with 100 μl of 99% ethyl acetate containing 1% TFA followed by 5% acetonitrile containing 0.2% TFA and eluted with 60% acetonitrile containing 1% ammonium hydroxide then dried by vacuum centrifugation.

**Mass spectrometry.** Peptides were resuspended in 2% acetonitrile, 0.1% TFA and loaded onto a 50 cm × 75 μm inner diameter column packed in-house with 1.9 μm C18AQ particles (Dr Maisch GmbH HPLC) using Dionex nanoUHPLC. Peptides were separated using a linear gradient of 5–30% Buffer B over 180 min at 300 nl/min (Buffer A = 0.1% formic acid; Buffer B = 80% acetonitrile, 0.1% formic acid). The column was maintained at 50 °C using a PRSO-V1 ion-source (Sonation) coupled directly to an Orbitrap Fusion Lumos mass spectrometer (MS). A full-scan MS1 was measured at 60,000 resolution at 200 $m/z$ (350–1650 $m/z$; 50 ms injection time; 4e5 automatic gain control target) followed by data-dependent analysis of the most abundant precursor ions for MS/MS by HCD (1.4 $m/z$ isolation; 30 normalised collision energy; 15,000 resolution; 28 ms injection time, 1e5 automatic gain control target, 2 s fixed cycle time).

**Proteomic data analysis.** Mass spectrometry data were processed using Andromeda/MaxQuant (v1.6.2.2)[69] and searched against the mouse UniProt database (June, 2019) using all default settings with peptide spectral matches and protein false-discovery rate (FDR) set to 1%. First search mass tolerances were set to 20 ppm for MS1 and MS2 and following recalibration, a second search was performed with MS1 tolerance of 4.5 ppm and MS2 tolerance of 20 ppm. The data were searched with a maximum of 2 miss-cleavages, and methionine oxidation and protein N-terminus acetylation were set as variable modifications while carbamidomethylation of cysteine was set as a fixed modification. The MaxLFQ algorithm was enabled including match between runs with default parameters[70]. Data were post-processed via Log2 transformation and median normalisation.

**Protein extraction and western blotting.** Mouse and human skeletal muscles were lysed in homogenisation buffer (50 mM Tris-HCL, 1 mM EDTA, 1% (vol/vol), Triton X-100, 0.1% (vol/vol) 2-mecaptoethanol; pH 7.5) supplemented with protease and phosphatase inhibitor cocktail. Protein concentrations were determined using the bicinchoninic acid (BCA®) assay as per manufacturer's instructions (Thermo Fisher Scientific). Proteins were separated and transferred to PVDF-FL membranes (Bio-Rad) that were air-dried at room temperature (RT) for 15 min, before re-hydration in absolute methanol and blocking for 1 h at RT in 5% non-fat milk that was diluted in phosphate buffered saline. Membranes were incubated overnight with primary antibodies, followed by incubation for 1 h at RT with HRP-conjugated secondary antibodies and detection using ECL detection reagent (GE Healthcare life sciences). Primary antibodies used were total YAP (#14074) and total IDH2 (#56439); all 1/1000 dilution from Cell Signalling Technologies and GAPDH (#SC-32233); 1/20,000 dilution from Santa-Cruz Biotechnology. Secondary antibodies used were goat anti-rabbit (#1706516, Bio-Rad technologies, 1/5000 dilution) or goat anti-mouse (#1706515, Bio-Rad technologies, 1/5000 dilution). Densitometry was performed using ImageJ software v1.53c (http://rsb.info.nih.gov/ij/index.html). Total protein levels as assessed by Ponceau S staining or Stain-free methodology (Bio-Rad) of the membrane following transfer was used for normalisation of loading for all mouse experiments. Human samples were normalised to the levels of GAPDH.

**Histological assessment.** Following surgical excision, muscles were quickly blotted, weighed, and transversely sectioned, so that a transverse section of each muscle could be embedded in cryomolds containing Optimum Cutting Temperature (Grale Scientific, Victoria, Australia), frozen in liquid nitrogen-cooled isopentane then stored at −80 °C. Haematoxylin and Eosin (H and E) staining was performed as per manufacturer's protocols (Trajan). Myofiber area was determined following labelling of tissue sections with Wheat-germ agglutinin conjugated to Alexa-fluor594 (W11262, Thermo Fisher Scientific). Succinate dehydrogenase (SDH) activity was assessed by histochemical reaction of unfixed, frozen tissue sections[71]. TUNEL labelling of muscle cryosections was performed using a TACS TdT in situ Apoptosis detection kit (BioScientific, Australia) following manufacturer's instructions. ROS was assessed in sections that were fixed in 4% PFA diluted in PBS for 10 min using DHE following manufacturer's instructions (Sigma; 37291). All labelling was performed on 8 μm thick cryosections and imaged on a Ziess Axio D1 microscope.

Inguinal WAT immunohistochemistry was performed paraffin embedded formalin fixed inguinal fat. Paraffin embed tissues were sectioned at 8 μm. For detection of Ucp-1, sections were subjected to antigen retrieval in citrate acid buffer [10 mM Sodium Citrate, 0.05% (v/v) Tween 20, pH 6.0] at 85 c for 10 min. Sections were incubated at RT for 1 h in blocking buffer [0.1 M phosphate buffer, 0.2% (v/v) Triton X-100, 10% (v/v) normal goat serum (Sigma, St. Louis, MO); and then

overnight at 4c in rabbit anti-Ucp-1 (1/1000; ab10983, Abcam, Cambridge, UK), in blocking buffer. After washing with PBS, sections were incubated with goat anti-rabbit biotinylated Antibody (1/500, Vector Laboratories, CA, BA-1000) in blocking buffer for 2 h at RT. Ucp-1 antibody binding was visualised using rabbit IgG VECTORSTAIN ABC Elite and DAB (3,30-diaminobenzidine) Peroxidase Substrate Kits (Vector Laboratories, UK). Brightness and contrast have been adjusted to aid visualisation. Images were captured from 3 nonoverlapping fields of view with a representative image shown.

**Assessment of fatty acid metabolism in intact EDL muscles.** Fatty acid metabolism was assessed in intact (tendon-to-tendon isolated) EDL muscle examined 14 days after administration of AAV6:lacZ-shRNA or AAV6:Yap-shRNA[71]. Muscle was incubated for 2 h in warmed (37 °C), pre-gassed (95% $O_2$–5% $CO_2$) low-glucose DMEM containing 2% fatty acid-free BSA, 500 µmol/L oleic acid and 1 µCi/mL $1-^{14}$C-oleic acid (Perkin Elmer, Australia). Media was acidified in 1 mol/L perchloric acid, $CO_2$ captured in 1 mol/L NaOH, and radioactivity counted using a liquid-scintillation counter (TRI-CARB 4910TR 110 V Liquid-Scintillation Counter, Perkin Elmer). Tissue was homogenised in 2:1 (v/v) chloroform:methanol, phase separation initiated by addition of 0.9% NaCl, the aqueous layer used for assessment of acid-soluble metabolites and the lipid layer for assessment of TAG incorporation.

**Assessment of mitochondrial respiratory capacity in isolated mitochondria.** Isolated mitochondrial oxygen consumption was assessed via two assays in an XF-e96 Seahorse Bioanalyser (Agilent). Mitochondria from skeletal muscle were isolated via differential centrifugation, mitochondrial total protein measured and 5 ug per well of mitochondria seeded into an XF-e96 cell culture plate. Mitochondria were adhered to the plate via centrifugation of the plate (2000 × g for 10 min at 4 °C) in mitochondrial assay solution. In the first assay, Complex I-linked respiration was assessed in the presence of 10 mM pyruvate and malate. Following basal oxygen consumption readings, ADP (State 3 phosphorylating respiration: 3 mM final concentration), Oligomycin (Complex V inhibitor—State 4 respiration: 2 µM), FCCP (uncoupler-stimulated respiration: 2 µM) and Antimycin A (complex III inhibitor: 2 µM) were sequentially injected. In the second assay (electron flow measurements), recordings were made in the presence of 10 mM pyruvate and malate and 4 µM FCCP in the media. Injections were as follows Rotenone (complex I inhibitor: 2 µM final concentration), Succinate (complex II substrate: 10 mM), Antimycin A (complex III inhibitor: 4 µM), and Ascorbate (10 mM) with N,N,N′,N′-Tetramethyl-p-phenylenediamine (TMPD) (1 mM) acting as cytochrome C/ complex IV substrate. Data were analysed using Wave (v2.6).

**Assessment of respiration in intact EDL muscles.** Respiration in saponin permeabilized EDL muscle fibres was measured using high-resolution respirometry (Oxygraph-2k; Oroboros Instruments, Innsbruck, Austria) at 37 °C with a stirrer at 750 rpm. C57BL/6 male mice aged 8 weeks were humanely euthanised by cervical dislocation and the EDL muscle rapidly excised and placed in ice-cold BIOPS solution (10 mM $Ca^{2+}$–EGTA buffer, 0.1 µM free $Ca^{2+}$, 20 mM imidazole, 20 mM taurine, 50 mM K-MES, 0.5 mM DTT, 6.56 mM $MgCl^2$, 5.77 mM ATP, 15 mM phosphocreatine, pH 7.1). The EDL muscle was mechanically separated using forceps, but kept intact from tendon to tendon, to generate a muscle fibre bundle of ~2 mg. The muscle bundle was permeabilised in BIOPS buffer containing Saponin (50 µg/ml) on ice with gentle agitation for 30 mins, followed by 3 × 7 min washes in MiRO5 media (0.4 mM EGTA, 3 mM $MgCl^2$, 5 mM KH2PO4, 0.3 mM DTT, 125 mM KCl, 20 mM HEPES). Muscle bundles were then transferred into the Oroboros chambers and equilibrated with MiR05 media supplemented with octanoylcarnitine (0.2 mM). Respiration was measured by sequential injections of ADP (5 mM), malate (2 mM; complex I), glutamate (10 mM; Complex I), succinate (10 mM; complex I + II), Cytochrome C (10 µM), ADP (5 mM), FCCP (0.5 µM steps × 3; maximal uncoupled), rotenone (0.5 µM; Complex I inhibitor), and antimycin A (2.5 µM; complex III inhibitor). Cytochrome C addition was used to ensure mitochondrial integrity. Measurements were normalised to wet weight per chamber and data was analysed using Datlab2 software (Oroboros Instruments v7.4).

**Body composition and metabolic caging analysis of mice.** Body mass was determined using laboratory scales (Mettler Toledo). Fat and lean mass was determined using quantitative magnetic resonance (4-in-1-500, EchoMRI™)[72]. Energy expenditure and total activity were determined using Comprehensive Lab Animals Monitoring System (Columbus Instruments) following manufacturer's protocols[72]. Energy Expenditure data was normalised to total lean mass determined the day prior to commencing metabolic cage studies.

**Determination of insulin and c-peptide levels.** Fasting insulin levels were measured in animals fasted for 5 h following indicated interventions by blood sampling via tail vein and determined by mouse insulin ELISA (Alpco) following manufacturer's instructions. C-Peptide levels were determined by mouse c-peptide ELISA (Alpco). Insulin secretion following glucose challenge was determined at 0 and at 15 min following oral gavage of glucose (2 mg/g body weight).

**Steady state-polar metabolomics and lipidomics.** Soleus muscles of C57BL/6J mice were treated with AAV6:lacZ-shRNA or AAV6:Yap-shRNA vectors and excised after 4 weeks for homogenisation in pre-chilled solutions on dry ice. The extraction of metabolites was initiated by the addition of 80:20 Methanol:Water containing 2 µM each of $^{13}$C-AMP, $^{13}$C-UMP, $^{13}$C Sorbitol, and $^{13}$C Valine to the already snap-frozen samples, as internal standards. Muscles were homogenised using a micro homogeniser connected to a pre-chilled pellet pestle. Samples were then vortexed for 30 s before being sonicated for 5 min in an ice bath. Samples were incubated for 5 min at 4 °C before being fractionated into two groups of 250 µL each for steady-state-polar metabolomics and lipidomics analyses. The 250 µL volume for steady state-polar metabolomics was thermomixed for 10 min at 4 °C, then centrifuged at 18928 × g for 10 min at 4 °C before injecting 7 µL into the Agilent hydrophilic interaction (HILIC) LC and high-resolution mass spectrometry (QTOF-6545). Metabolite peak calling and quality check was performed using QTOF MassHunter Quant software (Agilent). The remaining 250 µL of extract was used for lipidomic analysis following rigorous vortex agitation to resuspend the pellet. A 100 µL aliquot of this resuspended solution was used for subsequent lipid extraction. Lipids were extracted using 2:1 Methanol:Chloroform containing 2 µM each of internal standards of PC 19:0/19:0, PG 17:0/17:0, PE-d31, and TG-d5. Samples were then incubated in a Thermomix at 20 °C for 15 min, before centrifugation at 25200 × g at 4 °C for 10 min. The supernatant was evaporated in a rotational vacuum concentrator with nitrogen to complete dryness. Finally, the samples were reconstituted in 9:1 Butanol:Methanol before 2 µL injection into the Lipid Liquid Chromatography Mass Spectrometer (QQQ-6590, Agilent). Metabolite peak calling and quality check was performed using QQQ MassHunter Quant software (Agilent). Samples were normalised according to the soluble protein concentration as determined by DC Protein assay kit (Bio-Rad). Heatmaps were generated using MetaboAnalyst 4.0[73].

**Human muscle biopsies.** Human muscle biopsies were obtained from obese, skeletal muscle insulin-sensitive and insulin-resistant individuals[74]. Subjects were fasted overnight and subjected to a 6 h hyperinsulinemic-euglycemic clamp. A 2 h infusion of low-dose insulin (15 mU/m²/min) and a 2 h infusion of high-dose insulin (80 mU/m²/min) in the euglycemic state were performed and glucose infusion rates (GIR) were calculated over a 90–120 min period then normalised to subject FFM. Whole-body RQ was obtained using indirect calorimetry in the fasted state (Fasted RQ) prior to the clamp and during the final 30 min (glucose-infused RQ). ΔRQ (a measure of whole-body metabolic flexibility) was determined as the difference between the glucose-infused RQ and Fasted RQ (Parvo Medics True One). Skeletal muscle insulin sensitivity was determined separately for men and women. Study participants were assigned to the muscle insulin-sensitive group if GIR was in the upper tertile of the cohort, and to the insulin-resistant group when GIR fell in the lower two tertiles. Hepatic glucose output was determined during the low-dose insulin infusion state.

**Statistics and reproducibility.** Graphpad Prism v8 was used for statistical analysis and generation of figures except for ATAC/RNA-sequencing, Proteomics and Metabolomics/Lipidomics datasets that were processed as indicated in the relevant methods section. Images in Figs. 1A, E, 3A were generated using Biorender (https://app.biorender.com). Sample sizes for experiments with animals were determined using predicted size of effects from previous work with an alpha value of 0.05 and a beta value of 0.1. Animals were randomly assigned to experimental groups with data presented as mean ± S.E.M. assumed to be normally distributed and are representative of independent biological replicates as indicated in the appropriate figure legends. The investigators were blinded to allocation during experiments or to experimental outcome. Comparisons between changes from control muscles were made using the Student's unpaired, two-sided $t$ test for two comparisons. One-way analysis of variance (ANOVA) test or two-way ANOVA as appropriate was used to compare multiple conditions with the Tukey's post-hoc test used for comparisons between specific group means. Significant differences reported are $p < 0.05$. Transcriptomics datasets were corrected for multiple comparisons using the Benjamini–Hochberg FDR method using a false-discovery rate of 5%. Statistical significance was calculated by Student's unpaired, two-sided $t$ test. Human datasets are presented as mean ± S.E.M and are representative of a total of 54 individuals. Data for human subjects were not normally distributed and so correlations in human datasets were performed using a Spearman's rank-order correlation test. Spearman's correlation co-efficient ($r_s$) and $p$ value are reported in the appropriate figure legends.

**Reporting summary.** Further information on research design is available in the Nature Research Reporting Summary linked to this article.

## Data availability

ATAC-seq and RNA-seq data reported in this paper have been deposited in the NCBI Gene Expression Omnibus and are accessible through the GEO Series accession number GSE144138. The mass spectrometry proteomics data have been deposited to the ProteomeXchange Consortium via the PRIDE partner repository with the dataset identifier PXD024379. Source data are provided with this paper.

## Code availability

All code used in the analysis of data and generation of related figures is accessible via https://github.com/markziemann/watt2021_yap and https://evangelynsim.github.io/2021_UoM_Yap_shRNA_nuclei_RNAseq_ATACseq/

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

## Acknowledgements
We thank Stephanie Jansen and Prue O'Hare, AMREP animal services, Melbourne for technical assistance. We acknowledge Dr Daniel Chan, Garvan Institute, Sydney who collected human muscle biopsies used in this study and express our gratitude to the subjects that participated in the studies that collected muscle biopsies. We acknowledge Metabolomics Australia for their contribution to this work. We also acknowledge the use of Illumina sequencing at the Australian Genome Research Facility and the Victorian Clinical Genetics Service (and the support they receive from the Commonwealth of Australia). Figure schematics were generated using BioRender (https://app.biorender.com/). This work was supported by a Project grant from the Australian National Health and Medical Research Council (NHMRC) (awarded to K.F.H. and P.G.) and a Diabetes Australia general grant (awarded to K.I.W.). P.G., A.E.O., M.J.W. and K.F.H. are supported by Senior Research Fellowships, M.A.F. by a Principal Research Fellowship and M.K.M. by a career development fellowship (all from the NHMRC). A.G.C. is supported by an Investigator grant from the NHMRC and a future fellowship from the Australian Research Council. B.G.D. is supported by a National Heart Foundation of Australia Future Leader Fellowship (101789). The University of Melbourne, Monash University, Deakin University, the Murdoch Children's Research Institute and the Baker Heart and Diabetes Institute are supported in part by the Operational Infrastructure Support Programme of the Victorian Government.

## Author contributions
K.I.W., K.F.H. and P.G. designed the studies, K.I.W., D.C.H., M.Z., C.B.S., D.S-B., B.L.P., M.K.M., T.M.S., G.T.D., R.L-Y., H.Q., R.E.T., A.H., J.R.D., R.K. and S.T.B, performed experimental procedures, K.I.W., D.C.H., M.Z., C.B.S., M.K.M., D.S-B., T.M.S., B.L.P., R.K., A.H., S.T.B. and A.G.C. performed data analysis, A.E-O., J.R.G., B.G.D., M.J.W., M.A.F., A.G.C., E.R.P., K.F.H. and P.G., provided resources and supervision, K.I.W., P.G. wrote the original draft, all authors contributed to revisions and approved the final draft, K.I.W., K.F.H. and P.G. acquired funding.

## Competing interests
The authors declare no competing interests.
