## [Peer Review File · Nature Communications]

Reviewer Comments, first round –

Reviewer #1 (Remarks to the Author):

This well written manuscript reports that the protein concentration of the Hippo effector YAP in muscle is associated with fat oxidation and that modulating YAP levels alters fat oxidation. This not only has effects on muscle itself but also on other organs, suggesting either myokine effects, altered substrate availability or that the AAV-Yap constructs altered Yap levels not only in muscle. The study is a conceptual advance because generally Yap induces proliferation in mitotic cells and this typically induces the Warburg effect and thus carbohydrate metabolism. In contrast, study shows an increase of fat oxidation in post-mitotic muscle. The study does not suggest a mechanisms by which high-Yap-muscle affects white adipose tissue or the liver. Is it myokines? Perhaps the answer is in the omics data but even without that answer the new knowledge is considerable.

Major comments:

Major comment 1. Yap drives muscle fibre hypertrophy and fat metabolism. The authors should check whether there is evidence that hypertrophy or hypertrophy interventions (which of course not only depend on Yap) are increased with altered fat metabolism. Does resistance training increase fat metabolism? Do bodybuilders have increased levels of fat metabolism? Can the data help to explain why resistance training improves glycaemic control? Doing these analysis would help to fit this data into the bigger picture.

Major comment 2: Skeletal muscle is a highly heterogeneous tissue with type 1 fibres having higher levels of fat oxidation and type 2 fibres having more carbohydrate metabolism. In Figure 1, the researchers report the data for the slower soleus and faster TA in two figures. But is Yap higher in the soleus than TA which would be expected? The authors should also show the data from the paper from Marta Murgia, who has done single fibre proteomics in young and old type I and type II human muscle fibres. These data show higher YAP protein in slow than fast and young than old muscle fibres. You can also associate Yap protein levels with the levels of fat oxidation enzymes (see the supplementary Excel file of that study) they show higher levels of YAP in human type I than type II fibres and younger individuals have higher levels of Yap than older individuals: Gene names Slow younger Slow older Fast, 2A younger Fast, 2A older
YAP1 602.4 245.1 297.1 154.5

Major comment 2: Figure 4 shows nice physiological and biochemical data but the key question is whether the YAP was only increased in muscle by the AAV so that the subcutaneous fat and liver effects are mediated presumably by myokines. Which ones? Any hint in the RNAseq & proteomics data? Alternatively, the AAV-delivered YAP to many organs so that the interpretation is much more complex. Here, the authors should ideally show the results of assays that measure Yap in different muscles and organs. Also, is there really no YAP in the control gastrocnemii? The authors should show a blot with a longer exposure.

Minor comments:

Minor comment 1: Figure 3 This figure shows interesting data but how this relates to altered fat metabolism is not directly clear from the figure and could be illustrated more clearly. For example, the authors should name relevant at least some fat metabolism-related genes in figures 3B, C, E as well as H e.g. through colouring. H is unreadable. How 3G demonstrates a link to pyruvate metabolism, the TCA cycle and fat metabolism is unclear. Here, less data better explained would be more because this is the key figure where you try to convince the reader that YAP directly regulates a gene expression program related to fat metabolism.

Minor comment 2: # 145-147. The authors performed a polar metabolomics analysis but most lipids are apolar? This seems a bit surprising and the rationale could be explained further.

Minor comment 3: #270. Increase levels of CTGF occur in dystrophic muscles and there are some publications showing that the overexpression of CTGF causes a dystrophy/myopathy. Is there any evidence that these muscles have a dystrophy/myopathy phenotype?

Minor comment 4: # 300-301 Here, I would contrast the metabolic effects in postmitotic muscle with the metabolic effects in mitotic cells, as this highlights that these findings are specific to differentiated muscle.

Reviewer #2 (Remarks to the Author):

The present study attempted to establish a role for the Hippo pathway, particularly Yap, on regulation of skeletal muscle substrate selection and protection against lipotoxicity in the context of obesity. Their main findings are: 1) Yap protein levels are reduced in muscle biopsies from obese insulin resistant individuals; 2) AAV-shRNA-mediated decrease of Yap in skeletal muscles of mice increases incomplete FA oxidation rates, without changes in complete FA oxidation rates, potentially leading to lipotoxicity; 3) overexpression of Yap (AAV-Yap) in muscles of db/db mice decreases fat mass and liver steatosis. The presented findings are interesting, however some experiments are still needed to clarify how muscle Yap ameliorates the obese phenotype.

- The increased Yap expression in the soleus of db/db mice suggests a potential fiber-type specific impact on Yap expression upon nutrient oversupply. Is that specific to the db/db mouse? Further examining this aspect in the model of diet-induced obesity appears important.

- While characterizing the impact of Yap knockdown across muscles of different fiber types, the TA and GA muscles display a mixed fiber type composition, with low to very low % of type 1 fibers. It would be beneficial to have the EDL muscle included in the analysis, which contains primarily type 2b and 2x fibers, especially because the EDL was used in the characterization of FA oxidation rates.

- In Supplem. Fig. 3, what is the scale bar indicating? The fibers from the soleus cross-sections appear larger than the ones of the TA, which is strange. Also, please quantify the % of centrally nucleated fibers, which represent fibers undergoing degenerative/regenerative cycles thereby serving as an objective surrogate of spontaneous fiber damage. This appears to be relevant considering the increased DNA fragmentation nuclei and enhanced expression of inflammatory genes in muscles with deficient Yap levels.

- Yap knockdown led to increased incomplete FA oxidation, without changes in complete FA oxidation. This indicates that a higher FA influx to mitochondria was occurring in muscles deficient of Yap. Does Yap modulate the transport of AcylCoAs to mitochondria? Are Cpt1 levels increased? What about Malonyl CoA, which inhibit Cpt1?

- Less accessible regions of chromatin in Yap deficient muscles were enriched for transcription factor binding motifs of steroid hormone receptors and the TEAD transcription factors binding motifs. However, subsequent analyzes focused primarily on targets containing Tead-binding motifs. It appears that changes in genes responsive to steroid hormone receptors could have contributed to the phenotype and should be analyzed as well.

- How was the data in the Supplem. Fig. 7 C, D statistically tested? Are all conditions different?

- Determining metabolic rate in db/db mice overexpressing Yap appears necessary to understand the phenotype. A potential improved capacity for FA oxidation should also be tested. However, even if it is occurring, it is not sufficient to explain the reduced body fat gain which must result from a less positive energy balance in db/db + Yap vs. db/db + empty vector.

- Gtt's and Itt's should be performed in db/db + Yap mice vs. db/db + empty vector to understand if Yap overexpression improves insulin sensitivity.

- Fig 4K shows a clear increase in SDH activity in muscles overexpressing Yap. Are these changes occurring in parallel to MHC changes (i.e., decrease of MHC2b/x and increase of MHC2a and 1)? Are there changes in fiber types?

Reviewer #3 (Remarks to the Author):

Reviews Watt et al. "Yap regulates skeletal muscle fatty acid oxidation and adiposity in metabolic disease"

In the present manuscript the authors' Watt et al. identify a metabolic role of YAP in skeletal muscle that has functional relevance in whole-body metabolic homeostasis. The authors identify first a differential regulation of YAP levels in different models of metabolic disorders. They show that YAP levels are reduced in insulin-resistant patients compared to insulin-sensitive individuals and showed that fast-twitched skeletal muscle of diabetic mice (db/db) have also a reduction of YAP levels. The authors hypothesize that YAP has a metabolic role in skeletal muscle, to prove this hypothesis they generated loss and gain of function models in db/db mice using AAVs in db/bd mice. Using this approach, Watt et al. show that loss of YAP leads to decreased fatty acid oxidation and lipotoxicity. On the other hand, YAP overexpression reduces the accumulation of fat mass and hepatic steatosis. The authors perform in addition RNAseq, ATACseq, and proteomics to claim the role of YAP in programming metabolism and substrate utilization.

The role of YAP in metabolism has been previously addressed and reviewed (Hyun Koo et al., Cell Metabolism 2018, Cox et al. Nature 2016, Santinon et al. Trends in Cell Biology 2016, Tharp et al. Cell Metabolism 2018 or Jeong et al JCI 2018), the authors could discuss previously known metabolic roles of YAP. The roles of YAP in metabolism seem to differ across tissues and its role in skeletal muscle metabolism remains elusive. Overall the manuscript shows conclusive and solid phenotypes. First the observation of the change in YAP levels in skeletal muscle of mice and humans. Then, the authors choose a smart strategy to manipulate YAP levels in skeletal muscle using AAVs, the results show different metabolic effects consistent with their initial observations. The strength of the paper is the observation of novel findings in YAP regulation of skeletal muscle in humans, that they can further investigate with mouse models. The strength of the models is that knockdown or overexpression YAP in db/db mice show opposite metabolic effects. The weakness of this manuscript resides however on a lack of molecular mechanisms to explain these effects. The authors perform several unbiased approaches, RNAseq, ATACseq, and proteomics which result to be only very descriptive. These approaches could have been used to rule out a more precise molecular mechanism.

The authors should provide additional mechanistic experiments regarding the molecular function of YAP in skeletal muscle cells. The authors should explain how YAP signaling mediators interplay in skeletal muscle and how they are affected upon YAP loss or of gain of function in skeletal muscle cells. In addition, the manuscript would benefit from a better understanding of the direct metabolic targets of YAP-Tead complexes. Which are key target genes by ChIP?

An interesting point is to decipher how YAP levels are reduced upon insulin resistance. How is this regulated at the molecular level? The C-terminal region of TAZ/YAP contains a serine-rich phospho-degron motif that, when phosphorylated, targets TAZ/YAP for ubiquitylation and proteasome-mediated degradation. The phosphorylation of S397 targets YAP for degradation. How is this affected in insulin resistance?

The phenotypes are solid and consistent however, a GTT and ITT upon loss and gain of function of YAP were not shown.

There are additional minor points that should be addressed:

- Fig Supl 1A should show to which group belong each number of patients in the WB.
- Fig 1F, Fig supl B-C should show loading control done by WB, GAPDH for example like was done

for Fig. suppl 1A.

- Provide a rationale on the statistical test used in Fig 1B. The sample size between the 2 groups is very different (double in insulin resistant). The distribution is not normal? It is not clear which statistical test was used in 1B.
- Fig 7C is not clear, seems like there is an overexpression of YAP in the liver upon viral infection, at least it is clear on 5 mice, the levels are clearly upregulated.
- The muscle levels of YAP upon viral infection should be shown by WB, what is the level of overexpression compared to the endogenous levels?

Very minor:

Line119: typo "predominately"

Point-by-Point response to Reviewers' comments

Reviewer 1:

This well written manuscript reports that the protein concentration of the Hippo effector YAP in muscle is associated with fat oxidation and that modulating YAP levels alters fat oxidation. This not only has effects on muscle itself but also on other organs, suggesting either myokine effects, altered substrate availability or that the AAV-Yap constructs altered Yap levels not only in muscle. The study is a conceptual advance because generally Yap induces proliferation in mitotic cells and this typically induces the Warburg effect and thus carbohydrate metabolism. In contrast, study shows an increase of fat oxidation in post-mitotic muscle. The study does not suggest a mechanisms by which high-Yap-muscle affects white adipose tissue or the liver. Is it myokines? Perhaps the answer is in the omics data but even without that answer the new knowledge is considerable.

Major comment 1. Yap drives muscle fibre hypertrophy and fat metabolism. The authors should check whether there is evidence that hypertrophy or hypertrophy interventions (which of course not only depend on Yap) are increased with altered fat metabolism. Does resistance training increase fat metabolism? Do bodybuilders have increased levels of fat metabolism? Can the data help to explain why resistance training improves glycaemic control? Doing these analysis would help to fit this data into the bigger picture.

In our Discussion (paragraph 2), we highlighted evidence that links fat metabolism to muscle hypertrophy in non-obese states. In contrast, impaired anabolic signalling e.g. via the AKT/mTOR pathway occurs in obesity (*Sitnick et al. J Physiol 2009*). Our findings in *db/db* mice (Fig 5) are consistent with this model, since we observed marked improvements in metabolism without any change in lean mass/muscle mass. We conclude that the anabolic and metabolic roles of YAP in skeletal muscle are distinct.

Resistance training has been reported to have no effect on 24-hr fat oxidation in non-obese young adult males and females (Melanson et al 2002, 2005). In older women, one study has shown resistance training causes alterations in substrate utilisation where fat oxidation was increased, while carbohydrate oxidation was decreased (Treuth et al 1995). Interpretation of this outcome is challenging due to factors such as intensity and duration of exercise, training status, the age of the subject and the dietary and disease status of the subject. Our studies were focused on the significance of YAP in relation to metabolic disease and obesity, for which greater insight into disease mechanisms, and the development of novel therapeutic strategies is urgently required. As such, our experiments did not assess the impact of exercise on YAP activity or metabolic function and so we are unable to make comments in relation to this question. We agree with the Reviewer that this could be an interesting avenue for further study to examine potential interactions between the Hippo pathway, anabolic signalling, and fatty acid metabolism. However, to comprehensively examine the significance of YAP in relation to different modalities of exercise in both animal models and humans with consideration for age and disease status is a very different question, and of such scope that it deserves to be the focus of a dedicated manuscript.

Major comment 2: Skeletal muscle is a highly heterogeneous tissue with type 1 fibres having higher levels of fat oxidation and type 2 fibres having more carbohydrate metabolism. In Figure 1, the researchers report the data for the slower soleus and faster TA in two figures. But is Yap higher in the soleus than TA which would be expected? The authors should also show the data from the paper from Marta Murgia, who has done single fibre proteomics in young and old type I and type II human muscle fibres. These data show higher YAP protein in slow than fast and young than old muscle fibres. You can also associate Yap protein levels with the levels of fat oxidation enzymes (see the supplementary Excel file of that study) they show higher levels of YAP in human type I than type II fibres and younger individuals have higher levels of Yap than older individuals:

Gene names Slow younger Slow older Fast, 2A younger Fast, 2A older YAP1 602.4 245.1 297.1 154.5

We have previously reported YAP levels are higher in soleus vs TA muscles (Watt KI et al 2010). The findings of Murgia et al 2017 are consistent with our observations (demonstrate 50 % higher YAP levels in type 1 vs IIa) but with greater resolution (single fibres vs whole muscle lysates). The Murgia et al paper was referenced in our Introduction where we highlight the known inputs that impact YAP abundance/ activity.

Major comment 3: Figure 4 shows nice physiological and biochemical data but the key question is whether the YAP was only increased in muscle by the AAV so that the subcutaneous fat and liver effects are mediated presumably by myokines. Which ones? Any hint in the RNAseq & proteomics data? Alternatively, the AAV-delivered YAP to many organs so that the interpretation is much more complex. Here, the authors should ideally show the results of assays that measure Yap in different muscles and organs. Also, is there really no YAP in the control gastrocnemii? The authors should show a blot with a longer exposure.

In the original manuscript, we provided a blot of the FLAG tag (the over-expressed protein is a N-terminal FLAG-YAP fusion). As such, there was no protein band expected in the *db/db* empty vector or *db/+* lanes. To improve clarity for the reader, we have included Western blots of total YAP levels from gastrocnemius muscles and livers (Supp Fig 9B, C). We have also included qPCR data that demonstrate that the adipose tissue was not transduced by AAV vectors (Supp Fig 11F). Protocols for delivery of AAV6 vectors have been shown to achieve highly selective striated muscle transduction by our group (e.g. Winbanks et al 2012, 2016, Davey et al 2020) and others.

While we can't rule out myokine-mediated actions at this time, our demonstration that *db/db* mice administered AAV:Yap have higher energy expenditure during the active phase (Fig 5) provides a plausible explanation for their reduced adiposity and steatosis.

Minor comments:

Minor comment 1: Figure 3 This figure shows interesting data but how this relates to altered fat metabolism is not directly clear from the figure and could be illustrated more clearly. For example, the authors should name relevant at least some fat metabolism-related genes in figures 3B, C, E as well as H e.g. through colouring. H is unreadable. How 3G demonstrates a link to pyruvate metabolism, the TCA cycle and fat metabolism is unclear. Here, less data better explained would be more because this is the key figure where you try to convince the reader that YAP directly regulates a gene expression program related to fat metabolism.

We thank the reviewer for this comment. Following our latest analyses, we have extensively revised the presentation of the ATAC/RNA-seq expts in Fig 3 to provide a clearer and more effective summary of the data for readers.

Minor comment 2: # 145-147. The authors performed a polar metabolomics analysis but most lipids are apolar? This seems a bit surprising and the rationale could be explained further.

In Figure 2, we performed analyses of both polar metabolomics and lipidomics. We have now clarified the text of the revised manuscript to explain these aspects more clearly.

Minor comment 3: #270. Increase levels of CTGF occur in dystrophic muscles and there are some publications showing that the overexpression of CTGF causes a dystrophy/myopathy. Is there any evidence that these muscles have a dystrophy/myopathy phenotype?

We see no evidence of loss of muscle fibre integrity, or notable prevalence of centrally located nuclei by histological assays, as might be expected of a dystrophy phenotype. We also have not observed evidence of genes/proteins associated with muscle regeneration/degeneration in our transcriptomic or proteomic analyses either. CTGF is an established YAP target gene (including in muscle see Watt et al 2015 and Supp Fig 10) so is increased when YAP is overexpressed. Our approach focusses on understanding the basal role of Yap by a knockdown strategy. Consequently, we would not expect CTGF to increase in muscles where Yap has been knocked down. Our previous work (Watt et al 2015), and that of Judson et al 2013, show that YAP,

like CTGF, can cause a degenerative phenotype, but only when overexpressed at supra-physiological levels (>12 fold).

Minor comment 4: # 300-301 Here, I would contrast the metabolic effects in postmitotic muscle with the metabolic effects in mitotic cells, as this highlights that these findings are specific to differentiated muscle.

As requested, we have highlighted the differences between our model system and mitotically active cells in the Discussion (please see lines 331-342).

[continued on next page]

Reviewer 2:

The present study attempted to establish a role for the Hippo pathway, particularly Yap, on regulation of skeletal muscle substrate selection and protection against lipotoxicity in the context of obesity. Their main findings are: 1) Yap protein levels are reduced in muscle biopsies from obese insulin resistant individuals; 2) AAV-shRNA-mediated decrease of Yap in skeletal muscles of mice increases incomplete FA oxidation rates, without changes in complete FA oxidation rates, potentially leading to lipotoxicity; 3) overexpression of Yap (AAV-Yap) in muscles of db/db mice decreases fat mass and liver steatosis. The presented findings are interesting, however some experiments are still needed to clarify how muscle Yap ameliorates the obese phenotype.

We thank the Reviewer for their consideration of our work. We have sought to address their specific comments by completing new experiments and providing new data, as described below.

- The increased Yap expression in the soleus of db/db mice suggests a potential fiber-type specific impact on Yap expression upon nutrient oversupply. Is that specific to the db/db mouse? Further examining this aspect in the model of diet-induced obesity appears important.

To address this point, we have performed chow vs high-fat diet (DIO, 43 % fat for 6 weeks) feeding studies in mice and assessed YAP levels in the TA and *soleus* muscles (Supp Fig 2B). In contrast with our findings in a genetic-induced model of obesity (*db/db* mice) and the muscles of obese insulin-resistant humans, we found no difference in the levels of YAP between mice receiving different diets. The findings suggest that the differences in YAP levels in mice are related to the conditions present in the hyperphagic *db/db* strain. Importantly, our findings in *db/db* mice are consistent with the changes we observe in humans (Fig 1) providing relevance for these observations.

- While characterizing the impact of Yap knockdown across muscles of different fiber types, the TA and GA muscles display a mixed fiber type composition, with low to very low % of type 1 fibers. It would be beneficial to have the EDL muscle included in the analysis, which contains primarily type 2b and 2x fibers, especially because the EDL was used in the characterization of FA oxidation rates.

Adult mice of the C57BL/6 strain have similar fibre type composition between EDL and TA muscles (see "*Skeletal muscle fiber types in C57BL6J mice.*" Augusto et al 2004). Our comparison of a glycolytic (TA), mixed (Gastroc) and largely oxidative (soleus) muscle allows us to address this research question. The EDL muscle also displays a reduction in mass following Yap shRNA treatment as expected. However, the assessment of fatty acid oxidation is presented normalised to wet muscle mass and so has accounted for differences in mass when considering effects on substrate metabolism.

- In Supplem. Fig. 3, what is the scale bar indicating? The fibers from the soleus cross-sections appear larger than the ones of the TA, which is strange. Also, please quantify the % of centrally nucleated fibers, which represent fibers undergoing degenerative/regenerative cycles thereby serving as an objective surrogate of spontaneous fiber damage. This appears to be relevant considering the increased DNA fragmentation nuclei and enhanced expression of inflammatory genes in muscles with deficient Yap levels.

As requested, we have now provided additional details on the scale bar sizes in the revised Legends.

The numbers of centrally located nuclei are very low (<10 per muscle) and consistent between control and experimental conditions. In our previous study, we describe how degenerative pathology due to YAP over-expression only occurs at supra-physiological levels (>12 fold over current studies). We see no evidence of degenerative pathology with Yap knockdown, which is consistent with our previous work (Watt et al 2015).

- Yap knockdown led to increased incomplete FA oxidation, without changes in complete FA oxidation. This indicates that a higher FA influx to mitochondria was occurring in muscles deficient of Yap. Does Yap modulate the transport of AcylCoAs to mitochondria? Are Cpt1 levels increased? What about Malonyl CoA, which inhibit Cpt1?

We re-examined our data for effects on Cpt1. Although Cpt1 was detected in the proteomics dataset, no differences were observed between conditions.

To address this point, we have reanalysed our 'omics data. Comparing our transcriptomics datasets to published ChIP-seq studies allowed us to identify regulatory elements (enhancers and promoters) where chromatin accessibility and associated target gene expression were decreased following Yap knockdown. These new analyses identified several metabolism related genes including the TCA cycle enzyme, Idh2.

Idh2 was prioritised for follow-up investigation because our transcriptomics/proteomics/metabolomics comparisons all pointed to alterations in the TCA cycle as a feature of muscles treated with Yap shRNA. Idh2 is required for the generation of alpha-ketoglutarate from Acetyl-CoA>citrate in the TCA cycle. Yap knockdown resulted in increased Acetyl-CoA and reduced alpha-ketoglutarate levels (metabolomics), as well as increased levels of Suclg2 (the enzyme that metabolises alpha-ketoglutarate to succinate) and decreased levels of Pdk4 and Acadsb (Proteomics). Given the accumulation of TAG (lipidomics) and elevated rates of incomplete fatty acid oxidation in muscles expressing YAP shRNA, we reasoned that Yap must function to regulate the expression of genes that control substrate utilisation and flux through the TCA cycle. Since Idh2 is a rate limiting enzyme in this essential biological process, we assessed the impact of overexpression of this enzyme in muscles following Yap knockdown. Supporting this hypothesis, Idh2 overexpression prevented the impaired fatty acid oxidation phenotype normally observed following Yap knockdown. These latest findings provide important new mechanistic insights into the actions of Yap in adult skeletal muscle, and are now included in the revised Fig 4, with description in lines 232-261 of the Results.

- Less accessible regions of chromatin in Yap deficient muscles were enriched for transcription factor binding motifs of steroid hormone receptors and the TEAD transcription factors binding motifs. However, subsequent analyzes focused primarily on targets containing Tead-binding motifs. It appears that changes in genes responsive to steroid hormone receptors could have contributed to the phenotype and should be analyzed as well.

We sought to explore this angle further but did not find any logical candidates for follow up analysis using motif analysis. This is likely because the motif analysis (HOMER) is not based on experimentally validated data. As such, we accessed ChIP-seq datasets via the Cistrome database to perform the new integrated transcriptomics analysis described in Fig 4 which led to our identification of Idh2 as a functionally important downstream target of Yap in skeletal muscle. Our analysis does not rule out a potentially important role for steroid hormone receptors in muscle lacking Yap (the glucocorticoid receptor was in fact identified as downregulated in our transcriptomics analysis), but we are unable to determine a potential functional role for these receptors at this time.

- How was the data in the Supplem. Fig. 7 C, D statistically tested? Are all conditions different?

In the Methods, we have described the use of an ANOVA with Tukey's post-hoc test which is the appropriate statistical test to be used in this setting. The differences between groups have been highlighted in the relevant graphs.

- Determining metabolic rate in db/db mice overexpressing Yap appears necessary to understand the phenotype. A potential improved capacity for FA oxidation should also be tested. However, even if it is occurring, it is not sufficient to explain the reduced body fat gain which must result from a less positive energy balance in db/db + Yap vs. db/db + empty vector.

We agree with the Reviewer and so performed metabolic cage studies to address this point, which we have presented in the revised manuscript. These experiments show that, in addition to a restoration of post-prandial insulin secretion, obese mice with elevated levels of Yap in skeletal muscle display higher energy expenditure in the active (dark) phase. We attempted to measure fatty acid oxidation in these mice at

experimental endpoint but encountered technical difficulties that hampered our ability to make such conclusions. The new metabolic data are presented in Fig. 5 G and H of the revised manuscript.

- Gtt's and Itt's should be performed in db/db + Yap mice vs. db/db + empty vector to understand if Yap overexpression improves insulin sensitivity.

We have included data from an oGTT showing that *db/db* mice treated with AAV6:Yap vs empty vector have similar glucose tolerance (Fig. 5F). We attempted to perform an Insulin tolerance test but encountered technical difficulties that hampered our ability to interpret our findings. While we were unable to test insulin tolerance experimentally, we highlight our data in insulin-resistant humans that shows YAP levels correlate with Insulin sensitivity as determined by hyperinsulinemic-euglycemic clamp (Fig. 1).

- Fig 4K shows a clear increase in SDH activity in muscles overexpressing Yap. Are these changes occurring in parallel to MHC changes (i.e., decrease of MHC2b/x and increase of MHC2a and 1)? Are there changes in fiber types?

We were unable to reliably perform this assay on the tissue sections from these samples currently and so are unable to confirm that the changes in SDH activity occur independently of changes in MHC content. However, our proteomics analysis of muscles treated with AAV6:Yap-shRNA shows only a modest increase in the abundance of 3 myosins (Myl1 (fast), Myl12b (non-muscle myosin) and Myh8 (fetal)) suggesting that marked differences in fibre type are unlikely following changes in Yap activity.

[continued on next page]

Reviewer 3:

In the present manuscript the authors' Watt et al. identify a metabolic role of YAP in skeletal muscle that has functional relevance in whole-body metabolic homeostasis. The authors identify first a differential regulation of YAP levels in different models of metabolic disorders. They show that YAP levels are reduced in insulin-resistant patients compared to insulin-sensitive individuals and showed that fast-twitched skeletal muscle of diabetic mice (db/db) have also a reduction of YAP levels. The authors hypothesize that YAP has a metabolic role in skeletal muscle, to prove this hypothesis they generated loss and gain of function models in db/db mice using AAVs in db/bd mice. Using this approach, Watt et al. show that loss of YAP leads to decreased fatty acid oxidation and lipotoxicity. On the other hand, YAP overexpression reduces the accumulation of fat mass and hepatic steatosis. The authors perform in addition RNAseq, ATACseq, and proteomics to claim the role of YAP in programming metabolism and substrate utilization.

The role of YAP in metabolism has been previously addressed and reviewed (Hyun Koo et al., *Cell Metabolism* 2018, Cox et al. *Nature* 2016, Santinon et al. *Trends in Cell Biology* 2016, Tharp et al. *Cell Metabolism* 2018 or Jeong et al. *JCI* 2018), the authors could discuss previously known metabolic roles of YAP. The roles of YAP in metabolism seem to differ across tissues and its role in skeletal muscle metabolism remains elusive. Overall the manuscript shows conclusive and solid phenotypes.

We thank the Reviewer for their consideration of our work. In response to this feedback, we have revised the manuscript to more clearly highlight the differences between our model system and mitotically active cells in the Discussion (please see lines 331-342).

First the observation of the change in YAP levels in skeletal muscle of mice and humans. Then, the authors choose a smart strategy to manipulate YAP levels in skeletal muscle using AAVs, the results show different metabolic effects consistent with their initial observations. The strength of the paper is the observation of novel findings in YAP regulation of skeletal muscle in humans, that they can further investigate with mouse models. The strength of the models is that knockdown or overexpression YAP in db/db mice show opposite metabolic effects. The weakness of this manuscript resides however on a lack of molecular mechanisms to explain these effects. The authors perform several unbiased approaches, RNAseq, ATACseq, and proteomics which result to be only very descriptive. These approaches could have been used to rule out a more precise molecular mechanism.

The authors should provide additional mechanistic experiments regarding the molecular function of YAP in skeletal muscle cells. The authors should explain how YAP signaling mediators interplay in skeletal muscle and how they are affected upon YAP loss or of gain of function in skeletal muscle cells. In addition, the manuscript would benefit from a better understanding of the direct metabolic targets of YAP-Tead complexes. Which are key target genes by CHIP?

We thank the reviewer for this feedback. To address this point, we have re-analysed our transcriptomics datasets including a new comparison to published ChIP-seq data. We present these findings in the new Fig 3 and 4. These new analyses allowed us to identify putative targets of Yap in adult skeletal muscle including *Idh2* which we demonstrate is functionally important for efficient fatty acid oxidation in adult skeletal muscle. The identification of *Idh2* as a novel regulator of Yap effects in adult skeletal muscle provides important insight with exciting possibilities for future research, which we hope readers will appreciate.

An interesting point is to decipher how YAP levels are reduced upon insulin resistance. How is this regulated at the molecular level? The C-terminal region of TAZ/YAP contains a serine-rich phospho-degron motif that, when phosphorylated, targets TAZ/YAP for ubiquitylation and proteasome-mediated degradation. The phosphorylation of S397 targets YAP for degradation. How is this affected in insulin resistance?

To support our observations that Yap levels are reduced in glycolytic human muscles and glycolytic muscles of *db/db* mice, we have provided qPCR data to show that *Yap* mRNA expression is not different between

db/db and *db/+* strains, demonstrating that the alteration in Yap abundance between these settings is via a post-transcriptional mechanism.

Ser397 is the residue on Yap isoform 1. We have used Yap isoform 2 (the more abundant isoform in muscle) in these studies where the residue is Ser381. We were not able to detect serine 381 phosphorylation of YAP at this site in muscle lysates *in vivo* at the timepoints examined. Further, we were also unable to detect endogenous levels of pLATS1/2 as a proxy of this mechanism in tissue lysates. This is a challenging experiment to interpret since it is not clear what the appropriate timepoint to measure such a change would be (i.e. a timepoint where the protein is phosphorylated but not yet degraded). Identifying the upstream signalling mechanisms that control YAP activity in skeletal muscle *in vivo* is an area of active research for our team.

-The phenotypes are solid and consistent however, a GTT and ITT upon loss and gain of function of YAP were not shown.

To support our observations in obese mice treated with AAV6:Yap, we have now included new evidence that mice administered AAV6:Yap exhibit increased energy expenditure during the active (dark) phase and notable browning of the adipose tissue (suggestive of increased thermogenic capacity). We believe that these additional findings provide a plausible explanation for how the treated mice have lower adiposity and steatosis. We show that these mice still have impaired glucose tolerance by GTT (although post-prandial Insulin secretion has been restored). We attempted to perform an ITT in these mice but were unable to achieve this due to technical issues. The new findings provide valuable insight into the metabolic effects of increasing skeletal muscle Yap in a setting of obesity-related metabolic disease.

There are additional minor points that should be addressed:

Fig Supl 1A should show to which group belong each number of patients in the WB.

As requested, we have amended this figure element to provide greater clarity for readers.

- Fig 1F, Fig supl B-C should show loading control done by WB, GAPDH for example like was done for Fig. suppl 1A.

-We have opted to normalise the data using total protein stains (Ponceau or Stain-free) as loading controls for all Westerns in the mouse, as this will provide a more accurate indicator of equal loading than another protein such as GAPDH (*"The use of total protein stains as loading controls: an alternative to high-abundance single protein controls in semi-quantitative immunoblotting."* Aldridge et al 2008). While we would have liked to have performed similar analyses for the human cohorts, these blots were already run using GAPDH to normalise, and the available sample was limited, preventing such follow up.

- Provide a rationale on the statistical test used in Fig 1B. The sample size between the 2 groups is very different (double in insulin resistant). The distribution is not normal? It is not clear which statistical test was used in 1B.

The data presented in analysed using a Spearman's rank-correlation test because the distribution was determined as not normal. These details are provided in the Methods section – "Statistics".

- Fig 7C is not clear, seems like there is an overexpression of YAP in the liver upon viral infection, at least it is clear on 5 mice, the levels are clearly upregulated.

This figure was incorrectly labelled and we thank the reviewer for drawing our attention to this. We have corrected the relevant figure which is now presented in the revised Supp Fig 9.

- The muscle levels of YAP upon viral infection should be shown by WB, what is the level of overexpression compared to the endogenous levels?

As requested, we have now provided blots of total YAP levels (Supp Fig 9).

Very minor:

Line119: typo “predominately”

As requested, we have amended our text to address this typo.

[End of Document]

Reviewer Comments, second round –

Reviewer #1 (Remarks to the Author):

The authors have satisfactorily addressed all my comments with the exception of comment 3. Figure 5 I-K shows white adipose browning as UCP1 is upregulated in white adipose tissue in the mice that overexpress YAP in the musculature. Such UCP1 expression and browning is typically associated with increased energy expenditure. The authors should again check their data for myokines that can induce browning (there is a lot of literature on this) or at least discuss myokines as a possible mechanism by which Yap modulation in muscle may affect white adipose tissue.

Apart from that in 1D the "delta" sign has not plotted correctly.

Reviewer #2 (Remarks to the Author):

No further concerns.
Congratulations on a well written manuscript.

Reviewer #3 (Remarks to the Author):

The authors have successfully addressed all the points and included others that improve significantly the quality of the manuscript.
Very good work.

Point-by-Point response to Reviewer's comments

Reviewer #1 (Remarks to the Author):

The authors have satisfactorily addressed all my comments with the exception of comment 3. Figure 5 I-K shows white adipose browning as UCP1 is upregulated in white adipose tissue in the mice that overexpress YAP in the musculature. Such UCP1 expression and browning is typically associated with increased energy expenditure. The authors should again check their data for myokines that can induce browning (there is a lot of literature on this) or at least discuss myokines as a possible mechanism by which Yap modulation in muscle may affect white adipose tissue.

Apart from that in 1D the "delta" sign has not plotted correctly.

We acknowledge that our data does not preclude a potential link between the metabolic effects of YAP and myokines. We have included reference to this potential mechanism in the discussion as suggested (lines 345-349).

We have fixed the "delta" sign in Fig 1D.

We thank the reviewer for their constructive comments during the review process.

Reviewer #2 (Remarks to the Author):

**No further concerns.
Congratulations on a well written manuscript.**

We thank the reviewer for their constructive comments during the review process.

Reviewer #3 (Remarks to the Author):

**The authors have successfully addressed all the points and included others that improve significantly the quality of the manuscript.
Very good work.**

We thank the reviewer for their constructive comments during the review process.

Editors comments

Please clarify the source and details of the human samples used in this study.

Human samples were generated previously and used initially in *Chen et al. Phenotypic Characterization of Insulin-Resistant and Insulin-Sensitive Obesity. J Clin Endocrinol Metab 100, 4082–4091 (2015)*. We have indicated this clearly in the Methods section and included the following statement in Methods>Ethical approval – “Human muscle biopsies were procured previously with ethical consent provided by the St Vincent’s Hospital Human Research Ethics Committee (HREC/10/SVH/133. Sydney, Australia).”

Please provide reviewer’s tokens for GSE144138 (ATAC/RNA-sequencing datasets)

Reviewer’s can access GEO accession GSE144138 using the following link: <https://www.ncbi.nlm.nih.gov/geo/query/acc.cgi?acc=GSE144138> and entering the token wbkтуimenfunfgl into the box.

Point-by-Point response to Reviewer’s comments

Reviewer 1:

This well written manuscript reports that the protein concentration of the Hippo effector YAP in muscle is associated with fat oxidation and that modulating YAP levels alters fat oxidation. This not only has effects on muscle itself but also on other organs, suggesting either myokine effects, altered substrate availability or that the AAV-Yap constructs altered Yap levels not only in muscle. The study is a conceptual advance because generally Yap induces proliferation in mitotic cells and this typically induces the Warburg effect and thus carbohydrate metabolism. In contrast, study shows an increase of fat oxidation in post-mitotic muscle. The study does not suggest a mechanisms by which high-Yap-muscle affects white adipose tissue or the liver. Is it myokines? Perhaps the answer is in the omics data but even without that answer the new knowledge is considerable.

Major comment 1. Yap drives muscle fibre hypertrophy and fat metabolism. The authors should check whether there is evidence that hypertrophy or hypertrophy interventions (which of course not only depend on Yap) are increased with altered fat metabolism. Does resistance training increase fat metabolism? Do bodybuilders have increased levels of fat metabolism? Can the data help to explain why resistance training improves glycaemic control? Doing these analysis would help to fit this data into the bigger picture.

In our Discussion (paragraph 2), we highlighted evidence that links fat metabolism to muscle hypertrophy in non-obese states. In contrast, impaired anabolic signalling e.g. via the AKT/mTOR pathway occurs in obesity (*Sitnick et al. J Physiol 2009*). Our findings in *db/db* mice (Fig 5) are consistent with this model, since we observed marked improvements in metabolism without any change in lean mass/muscle mass. We conclude that the anabolic and metabolic roles of YAP in skeletal muscle are distinct.

Resistance training has been reported to have no effect on 24-hr fat oxidation in non-obese young adult males and females (Melanson et al 2002, 2005). In older women, one study has shown resistance training causes alterations in substrate utilisation where fat oxidation was increased, while carbohydrate oxidation was decreased (Treuth et al 1995). Interpretation of this outcome is

challenging due to factors such as intensity and duration of exercise, training status, the age of the subject and the dietary status of the subject. Our experimental design did not assess the impact of exercise training on YAP activity or metabolic function and so we are unable to make any comments in relation to this question. However, we agree with the reviewer that this would be a worthwhile avenue for further study that could provide more insight into the potential interactions between the Hippo pathway, anabolic signalling, and fatty acid metabolism.

Major comment 2: Skeletal muscle is a highly heterogeneous tissue with type 1 fibres having higher levels of fat oxidation and type 2 fibres having more carbohydrate metabolism. In Figure 1, the researchers report the data for the slower soleus and faster TA in two figures. But is Yap higher in the soleus than TA which would be expected? The authors should also show the data from the paper from Marta Murgia, who has done single fibre proteomics in young and old type I and type II human muscle fibres. These data show higher YAP protein in slow than fast and young than old muscle fibres. You can also associate Yap protein levels with the levels of fat oxidation enzymes (see the supplementary Excel file of that study) they show higher levels of YAP in human type I than type II fibres and younger individuals have higher levels of Yap than older individuals:

Gene names	Slow younger	Slow older	Fast, 2A younger	Fast, 2A older
YAP1	602.4	245.1	297.1	154.5

We have previously reported YAP levels are higher in soleus vs TA muscles (Watt KI et al 2010). The findings of Murgia et al 2017 are consistent with our observations (demonstrate 50 % higher YAP levels in type 1 vs IIa) but with greater resolution (single fibres vs whole muscle lysates). The Murgia et al paper was referenced in our Introduction where we highlight the known inputs that impact YAP abundance/ activity.

Major comment 3: Figure 4 shows nice physiological and biochemical data but the key question is whether the YAP was only increased in muscle by the AAV so that the subcutaneous fat and liver effects are mediated presumably by myokines. Which ones? Any hint in the RNAseq & proteomics data? Alternatively, the AAV-delivered YAP to many organs so that the interpretation is much more complex. Here, the authors should ideally show the results of assays that measure Yap in different muscles and organs. Also, is there really no YAP in the control gastrocnemii? The authors should show a blot with a longer exposure.

In the original manuscript, we provided a blot of the FLAG tag (the over-expressed protein is a N-terminal FLAG-YAP fusion). As such, there was no protein band expected in the *db/db* empty vector or *db/+* lanes. To improve clarity for the reader, we have included Western blots of total YAP levels from gastrocnemius muscles and livers (Supp Fig 9B, C). We have also included qPCR data that demonstrate that the adipose tissue was not transduced by AAV vectors (Supp Fig 11F). Protocols for delivery of AAV6 vectors have been shown to achieve highly selective striated muscle transduction by our group (e.g. Winbanks et al 2012, 2016, Davey et al 2020) and others.

While we can't rule out myokine-mediated actions at this time, our demonstration that *db/db* mice administered AAV:Yap have higher energy expenditure during the active phase (Fig 5) provides a plausible explanation for their reduced adiposity and steatosis.

Minor comments:

Minor comment 1: Figure 3 This figure shows interesting data but how this relates to altered fat metabolism is not directly clear from the figure and could be illustrated more clearly. For example, the authors should name relevant at least some fat metabolism-related genes in figures 3B, C, E as well as H e.g. through colouring. H is unreadable. How 3G demonstrates a link to pyruvate

metabolism, the TCA cycle and fat metabolism is unclear. Here, less data better explained would be more because this is the key figure where you try to convince the reader that YAP directly regulates a gene expression program related to fat metabolism.

We thank the reviewer for this comment and have extensively revised the presentation of the ATAC/RNA-seq expts in Fig 3 following our updated analysis to provide a clearer and more effective summary of the data for the reader.

Minor comment 2: # 145-147. The authors performed a polar metabolomics analysis but most lipids are apolar? This seems a bit surprising and the rationale could be explained further.

In Figure 2, we performed analyses of both polar metabolomics and lipidomics. We have now clarified the text of the revised manuscript to explain these aspects more clearly.

Minor comment 3: #270. Increase levels of CTGF occur in dystrophic muscles and there are some publications showing that the overexpression of CTGF causes a dystrophy/myopathy. Is there any evidence that these muscles have a dystrophy/myopathy phenotype?

We see no evidence of loss of muscle fibre integrity, or notable prevalence of centrally located nuclei by histological assays, as might be expected of a dystrophy phenotype. We also have not observed evidence of genes/proteins associated with muscle regeneration/degeneration in our transcriptomic or proteomic analyses either. CTGF is an established YAP target gene (including in muscle see Watt et al 2015 and Supp Fig 10) so is increased when YAP is overexpressed. Our approach focusses on understanding the basal role of Yap by a knockdown strategy. Consequently, we would not expect CTGF to increase in muscles where Yap has been knocked down. Our previous work (Watt et al 2015), and that of Judson et al 2013, show that YAP, like CTGF, can cause a degenerative phenotype, but only when overexpressed at supra-physiological levels (>12 fold).

Minor comment 4: # 300-301 Here, I would contrast the metabolic effects in postmitotic muscle with the metabolic effects in mitotic cells, as this highlights that these findings are specific to differentiated muscle.

We highlight the differences between our model system and mitotically active cells exist in the discussion (see lines 331-342).

Reviewer 2:

The present study attempted to establish a role for the Hippo pathway, particularly Yap, on regulation of skeletal muscle substrate selection and protection against lipotoxicity in the context of obesity. Their main findings are: 1) Yap protein levels are reduced in muscle biopsies from obese insulin resistant individuals; 2) AAV-shRNA-mediated decrease of Yap in skeletal muscles of mice increases incomplete FA oxidation rates, without changes in complete FA oxidation rates, potentially leading to lipotoxicity; 3) overexpression of Yap (AAV-Yap) in muscles of db/db mice decreases fat mass and liver steatosis. The presented findings are interesting, however some experiments are still needed to clarify how muscle Yap ameliorates the obese phenotype.

- The increased Yap expression in the soleus of db/db mice suggests a potential fiber-type specific impact on Yap expression upon nutrient oversupply. Is that specific to the db/db mouse? Further examining this aspect in the model of diet-induced obesity appears important.

To address this point, we have performed chow vs high-fat diet (DIO, 43 % fat for 6 weeks) feeding studies and assessed YAP levels in the TA and *soleus* muscles (Supp Fig 2B). In contrast with our findings in a genetic-induced model of obesity (*db/db* mice) and the muscles of obese insulin-resistant humans, we found no difference in the levels of YAP between conditions. The findings suggesting that the differences in YAP levels in mice are specific to *db/db* strain. Importantly, our findings in *db/db* mice are consistent with the changes we observe in humans (Fig 1) providing relevance for these observations.

- While characterizing the impact of Yap knockdown across muscles of different fiber types, the TA and GA muscles display a mixed fiber type composition, with low to very low % of type 1 fibers. It would be beneficial to have the EDL muscle included in the analysis, which contains primarily type 2b and 2x fibers, especially because the EDL was used in the characterization of FA oxidation rates.

Adult mice of the C57BL/6 strain have similar fibre type composition between EDL and TA muscles (see "*Skeletal muscle fiber types in C57BL6J mice.*" Augusto et al 2004). Our comparison of a glycolytic (TA), mixed (Gastroc) and largely oxidative (soleus) muscle allows us to address this research question. The EDL muscle also displays a reduction in mass following Yap shRNA treatment as expected. However, the assessment of fatty acid oxidation is presented normalised to wet muscle mass and so has accounted for differences in mass when considering effects on substrate metabolism.

- In Supplem. Fig. 3, what is the scale bar indicating? The fibers from the soleus cross-sections appear larger than the ones of the TA, which is strange. Also, please quantify the % of centrally nucleated fibers, which represent fibers undergoing degenerative/regenerative cycles thereby serving as an objective surrogate of spontaneous fiber damage. This appears to be relevant considering the increased DNA fragmentation nuclei and enhanced expression of inflammatory genes in muscles with deficient Yap levels.

We have now provided additional details on the scale bar sizes in the revised Legends.

The numbers of centrally located nuclei are very low (<10 per muscle) and consistent between control and experimental conditions. In our previous study, we describe how degenerative pathology due to YAP over-expression only occurs at supra-physiological levels (>12 fold over current studies). We see no evidence of degenerative pathology with Yap knockdown, which is consistent with our previous work (Watt et al 2015).

- Yap knockdown led to increased incomplete FA oxidation, without changes in complete FA oxidation. This indicates that a higher FA influx to mitochondria was occurring in muscles deficient of Yap. Does Yap modulate the transport of AcylCoAs to mitochondria? Are Cpt1 levels increased? What about Malonyl CoA, which inhibit Cpt1?

To address this point, we have reanalysed our 'omics data. Comparing our transcriptomics datasets to published ChIP-seq studies allowed us to identify regulatory elements (enhancers and promoters) where chromatin accessibility and associated target gene expression were decreased following Yap knockdown. These new analyses identified several metabolism related genes including the TCA cycle enzyme, *Idh2*.

Idh2 was prioritised for follow-up investigation because our transcriptomics/proteomics/metabolomics comparisons all pointed to alterations in the TCA cycle as a feature of muscles treated with Yap shRNA. *Idh2* is required for the generation of alpha-ketoglutarate from Acetyl-CoA>citrate in the TCA cycle. Yap knockdown resulted in increased Acetyl-

CoA and reduced alpha-ketoglutarate levels (metabolomics), as well as increased levels of *Suclg2* (the enzyme that metabolises alpha-ketoglutarate to succinate) and decreased levels of *Pdk4* and *Acadsb* (Proteomics). Given the accumulation of TAG (lipidomics) and elevated rates of incomplete fatty acid oxidation in muscles expressing YAP shRNA, we reasoned that Yap must function to regulate the expression of genes that control substrate utilisation and flux through the TCA cycle. Since *Idh2* is a rate limiting enzyme in this essential biological process, we assessed the impact of overexpression of this enzyme in muscles following Yap knockdown. Supporting this hypothesis, *Idh2* overexpression prevented the impaired fatty acid oxidation phenotype normally observed following Yap knockdown. We have included these new data in the revised Fig 4, with description in lines 232-261 of the Results.

We re-examined our data for effects on *Cpt1*. Although *Cpt1* was detected in the proteomics dataset, no differences were observed between conditions.

- Less accessible regions of chromatin in Yap deficient muscles were enriched for transcription factor binding motifs of steroid hormone receptors and the TEAD transcription factors binding motifs. However, subsequent analyzes focused primarily on targets containing Tead-binding motifs. It appears that changes in genes responsive to steroid hormone receptors could have contributed to the phenotype and should be analyzed as well.

We sought to explore this angle further but did not find any logical candidates for follow up analysis using motif analysis. This is likely because the motif analysis (HOMER) is not based on experimentally validated data. As such, we accessed CHIP-seq datasets via the Cistrome database to perform the new integrated transcriptomics analysis described in Fig 4 which led to our identification of *Idh2* as a functionally important downstream target of Yap in skeletal muscle. Our analysis does not rule out a potentially important role for steroid hormone receptors in muscle lacking Yap (the glucocorticoid receptor was in fact identified as downregulated in our transcriptomics analysis), but we are unable to determine a potential functional role for these receptors at this time.

- How was the data in the Supplem. Fig. 7 C, D statistically tested? Are all conditions different?

In the Methods, we have described the use of an ANOVA with Tukey's post-hoc test which is the appropriate statistical test to be used in this setting. The differences between groups have been highlighted in the relevant graphs.

- Determining metabolic rate in db/db mice overexpressing Yap appears necessary to understand the phenotype. A potential improved capacity for FA oxidation should also be tested. However, even if it is occurring, it is not sufficient to explain the reduced body fat gain which must result from a less positive energy balance in db/db + Yap vs. db/db + empty vector.

We agree with the Reviewer and so performed metabolic cage studies to address this point. These experiments show that, in addition to a restoration of post-prandial insulin secretion, obese mice with elevated levels of Yap in skeletal muscle display higher energy expenditure in the active (dark) phase. We attempted to measure fatty acid oxidation in these mice at experimental endpoint but encountered technical difficulties that hampered our ability to make such conclusions.

- Gtt's and Itt's should be performed in db/db + Yap mice vs. db/db + empty vector to understand if Yap overexpression improves insulin sensitivity.

We have included data from an oGTT showing that *db/db* mice treated with AAV6:Yap vs empty vector have similar glucose tolerance. We attempted to perform an Insulin tolerance test but

encountered technical difficulties that hampered our ability to interpret our findings. While we were unable to test insulin tolerance experimentally, we highlight our data in insulin-resistant humans that shows YAP levels correlate with Insulin sensitivity as determined by hyperinsulinemic-euglycemic clamp (Fig 1).

- Fig 4K shows a clear increase in SDH activity in muscles overexpressing Yap. Are these changes occurring in parallel to MHC changes (i.e., decrease of MHC2b/x and increase of MHC2a and 1)? Are there changes in fiber types?

We were unable to reliably perform this assay on the tissue sections from these samples currently and so are unable to confirm that the changes in SDH activity occur independently of changes in MHC content. However, our proteomics analysis of muscles treated with AAV6:Yap shRNA shows only a modest increase in the abundance of 3 myosins (My11 (fast), My12b (nonmuscle myosin) and Myh8 (fetal)) suggesting that marked differences in fibre type are unlikely following changes in Yap activity.

Reviewer 3:

In the present manuscript the authors' Watt et al. identify a metabolic role of YAP in skeletal muscle that has functional relevance in whole-body metabolic homeostasis. The authors identify first a differential regulation of YAP levels in different models of metabolic disorders. They show that YAP levels are reduced in insulin-resistant patients compared to insulin-sensitive individuals and showed that fast-twitched skeletal muscle of diabetic mice (db/db) have also a reduction of YAP levels. The authors hypothesize that YAP has a metabolic role in skeletal muscle, to prove this hypothesis they generated loss and gain of function models in db/db mice using AAVs in db/bd mice. Using this approach, Watt et al. show that loss of YAP leads to decreased fatty acid oxidation and lipotoxicity. On the other hand, YAP overexpression reduces the accumulation of fat mass and hepatic steatosis. The authors perform in addition RNAseq, ATACseq, and proteomics to claim the role of YAP in programming metabolism and substrate utilization.

The role of YAP in metabolism has been previously addressed and reviewed (Hyun Koo et al., Cell Metabolism 2018, Cox et al. Nature 2016, Santinon et al. Trends in Cell Biology 2016, Tharp et al. Cell Metabolism 2018 or Jeong et al JCI 2018), the authors could discuss previously known metabolic roles of YAP. The roles of YAP in metabolism seem to differ across tissues and its role in skeletal muscle metabolism remains elusive. Overall the manuscript shows conclusive and solid phenotypes.

We highlight the differences between our model system and mitotically active cells exist in the discussion (see lines 331-342).

First the observation of the change in YAP levels in skeletal muscle of mice and humans. Then, the authors choose a smart strategy to manipulate YAP levels in skeletal muscle using AAVs, the results show different metabolic effects consistent with their initial observations. The strength of the paper is the observation of novel findings in YAP regulation of skeletal muscle in humans, that they can further investigate with mouse models. The strength of the models is that knockdown or overexpression

YAP in db/db mice show opposite metabolic effects. The weakness of this manuscript resides however on a lack of molecular mechanisms to explain these effects. The authors perform several unbiased approaches, RNAseq, ATACseq, and proteomics which result to be only very descriptive. These approaches could have been used to rule out a more precise molecular mechanism.

The authors should provide additional mechanistic experiments regarding the molecular function of YAP in skeletal muscle cells. The authors should explain how YAP signaling mediators interplay in skeletal muscle and how they are affected upon YAP loss or of gain of function in skeletal muscle cells. In addition, the manuscript would benefit from a better understanding of the direct metabolic targets of YAP-Tead complexes. Which are key target genes by CHIP?

We thank the reviewer for this feedback. To address this point, we have re-analysed our transcriptomics datasets including a new comparison to published ChIP-seq data. We present these findings in the new Fig 3 and 4. These new analyses allowed us to identify putative targets of Yap in adult skeletal muscle including *Idh2* which we demonstrate is functionally important for efficient fatty acid oxidation in adult skeletal muscle.

An interesting point is to decipher how YAP levels are reduced upon insulin resistance. How is this regulated at the molecular level? The C-terminal region of TAZ/YAP contains a serine-rich phospho-degron motif that, when phosphorylated, targets TAZ/YAP for ubiquitylation and proteasome-mediated degradation. The phosphorylation of S397 targets YAP for degradation. How is this affected in insulin resistance?

To support our observations that Yap levels are reduced in glycolytic human muscles and glycolytic muscles of *db/db* mice, we have provided qPCR data to show that *Yap* mRNA expression is not different between *db/db* and *db/+* strains, demonstrating that the alteration in Yap abundance between these settings is via a post-transcriptional mechanism.

Ser397 is the residue on Yap isoform 1. We have used Yap isoform 2 (the more abundant isoform in muscle) in these studies where the residue is Ser381. We were not able to detect serine 381 phosphorylation of YAP at this site in muscle lysates *in vivo* at the timepoints examined. Further, we were also unable to detect endogenous levels of pLATS1/2 as a proxy of this mechanism in tissue lysates. This is a challenging experiment to interpret since it is not clear what the appropriate timepoint to measure such a change would be (i.e. a timepoint where the protein is phosphorylated but not yet degraded). Identifying the upstream signalling mechanisms that control YAP activity in skeletal muscle *in vivo* is an area of active research for our team.

-The phenotypes are solid and consistent however, a GTT and ITT upon loss and gain of function of YAP were not shown.

To support our observations in obese mice treated with AAV6:Yap, we have now included new evidence that mice administered AAV6:Yap exhibit increased energy expenditure during the active (dark) phase and notable browning of the adipose tissue (suggestive of increased thermogenic capacity). We believe that these additional findings provide a plausible explanation for how the treated mice have lower adiposity and steatosis. We show that these mice still have impaired glucose tolerance by GTT (although post-prandial Insulin secretion has been restored). We attempted to perform an ITT in these mice but were unable to achieve this due to technical issues. The new findings provide valuable insight into the metabolic effects of increasing skeletal muscle Yap in a setting of obesity-related metabolic disease.

**There are additional minor points that should be addressed:
Fig Supl 1A should show to which group belong each number of patients in the WB.**

We have amended this as requested by the reviewer.

- Fig 1F, Fig suppl B-C should show loading control done by WB, GAPDH for example like was done for Fig. suppl 1A.

-We have opted to normalise the data using total protein stains (Ponceau or Stain-free) as loading controls for all Westerns in the mouse, as this will provide a more accurate indicator of equal loading than another protein such as GAPDH (*"The use of total protein stains as loading controls: an alternative to high-abundance single protein controls in semi-quantitative immunoblotting."* Aldridge et al 2008). While we would have liked to have performed similar analyses for the human cohorts, these blots were already run using GAPDH to normalise, and the available sample was limited, preventing such follow up.

- Provide a rationale on the statistical test used in Fig 1B. The sample size between the 2 groups is very different (double in insulin resistant). The distribution is not normal? It is not clear which statistical test was used in 1B.

The data presented in analysed using a Spearman's rank-correlation test because the distribution was determined as not normal. These details are provided in the Methods section – "Statistics".

- Fig 7C is not clear, seems like there is an overexpression of YAP in the liver upon viral infection, at least it is clear on 5 mice, the levels are clearly upregulated.

This figure was incorrectly labelled and we thank the reviewer for drawing our attention to this. We have corrected the relevant figure which is now presented in the revised Supp Fig 9.

- The muscle levels of YAP upon viral infection should be shown by WB, what is the level of overexpression compared to the endogenous levels?

As requested, we have now provided blots of total YAP levels (Supp Fig 9).

Very minor:

Line119: typo "predominately"

We have amended our text to address this typo.

[End of Document]